


**Assimilation of citizen science data in snowpack modeling using a new**
**snow dataset: Community Snow Observations**
**Ryan L. Crumley[1], David F. Hill[2], Katreen Wikstrom Jones[3], Gabriel J. Wolken[3,4], Anthony A.**
**Arendt[5], Christina M. Aragon[1], Christopher Cosgrove[6], Community Snow Observations Participants[7]**
[1]Water Resources Science, Oregon State University, Corvallis, OR 97331, USA
[2]School of Civil and Construction Engineering, Oregon State University, Corvallis, OR 97331, USA
[3]Alaska Division of Geological and Geophysical Surveys, Fairbanks, AK 99709, USA
[4]International Arctic Research Center, University of Alaska Fairbanks, Fairbanks, AK 99775, USA
[5]University of Washington, Applied Physics Laboratory, WA 98105, USA
[6]Geography Department, Oregon State University, Corvallis, OR 97331, USA
[7]Citizen scientists participating in the project Community Snow Observations (CSO)
*Correspondence to*: Ryan L. Crumley (ryanlcrumley@gmail.com)
**Abstract.**
A physically-based snowpack evolution and redistribution model was used to test the effectiveness of assimilating crowd-sourced
measurements of snow depth by citizen scientists. The Community Snow Observations (CSO; communitysnowobs.org) project
gathers, stores, and distributes measurements of snow depth recorded by recreational users and snow professionals in high
mountain environments. These citizen science measurements are valuable since they come from terrain that is relatively under-
sampled and can offer *in-situ* snow information in locations where snow information is sparse or non-existent. The present study
investigates 1) the improvements to model performance when citizen science measurements are assimilated and 2) the number of
measurements necessary to obtain those improvements. Model performance is assessed by comparing time series of observed
(snow pillow) and modeled snow water equivalent values, by comparing spatially-distributed maps of observed (remotely sensed)
and modeled snow depth, and by comparing fieldwork results from within the study area. The results demonstrate that few citizen
science measurements are needed to obtain improvements in model performance and these improvements are found in 62% to 78%
of the ensemble simulations, depending on the model year. Model estimations of total water volume from a sub-region of the study
area also demonstrate improvements in accuracy after CSO measurements have been assimilated. These results suggest that even
modest measurement efforts by citizen scientists have the potential to improve efforts to model snowpack processes in high
mountain environments, with implications for water resource management and process-based snow modeling.
**1 Introduction**
The importance of snow in ecosystem function, in both human and natural systems, and in water resource management in western
North America cannot be overstated (Bales et al., 2006; Mankin et al., 2015; Viviroli et al., 2007). Internationally, more than a
billion people live in watersheds where snow is an integral part of the hydrologic system (Barnett et al., 2005). Snowpack dynamics
in mountainous, headwater catchments play an essential role connecting atmospheric processes and the hydrologic cycle with



downstream water users, agricultural systems, and municipal water systems (Fayad et al., 2017; Holko et al., 2011; Schneider et
al., 2013).
Information about snow distribution comes from many sources. First, there are snow datasets in the form of *in-situ* observations
of snowpack conditions, often observations of snow depth or snow water equivalent (SWE). In the United States of America (U.S.),
snow depth and SWE data are collected by the National Resources Conservation Service's (NRCS) Snow Telemetry (SNOTEL)
network using snow pillows and snow courses. Similar national *in-situ* snow observational networks exist in Europe, like the
MeteoSwiss and MeteoFrance programs that include snow depth, snowfall, and SWE datasets. For a comprehensive overview of
snow observations in Europe, including each program name, the location of observations, and agency websites, see the European
Snow Booklet (Haberkorn et al., 2019). Snow course information is also collected by state programs such as the California
Cooperative Snow Survey in the U.S. and, in the case of Canada, by provincial programs such as the British Columbia Snow
Survey. These *in-situ* snow observations provide critical information on snow conditions and snow distribution worldwide but vast
areas of snowpack remain unsampled.
To fill the observational gaps associated with point measurements, we often turn to snow information in the form of remote sensing
(RS) datasets, like the NASA-based Airborne Snow Observatory (Painter et al., 2016) that uses light detection and ranging
(LiDAR) in catchment-scale study areas. Other catchment-scale snow RS datasets are collected using unmanned aerial systems,
including high-elevation capable drones and balloon-based platforms in conjunction with structure-from-motion photogrammetry
(Buhler et al. 2016; Li et al., 2019). There are also RS datasets covering hemispheric and global scales, like the daily snow covered
area product from the MODIS satellite or the GlobSnow snow extent product from the European Space Agency (Hall & Riggs,
2016; Luojus et al., 2010).
Lastly, there are modeled snow datasets, like the Snow Data Assimilation project with a spatial extent that covers large portions of
North America (SNODAS; NOHRSC, 2004). There are physically-based snow models that produce snow information on
catchment- to hemisphere-scales, like iSnowBal, SnowModel, Alpine3D, PBSM, and SNOWPACK, among many others (Marks
et al., 1999; Liston & Elder, 2006a; Lehning et al, 2006; Pomeroy et al., 1993; Lehning et al., 1999). Studies that integrate all of
these types of snow information, *in-situ* observations, RS datasets, and process models, are becoming common in snow research
because they often produce the best results (Sturm et al., 2015).
Assimilation of data into process modeling is a strategy that seeks to incorporate measurements of environmental variables into
the model chain as a 'hybrid' approach to predicting modeled state variables (Carrassi et al., 2017; Kalnay et al., 2003). There are
many examples of data assimilation in the atmospheric sciences and weather prediction (Rabier et al., 2005), in weather reanalysis
products (Gelaro et al., 2017; Kalnay et al., 2003; Messinger et al., 2006; Saha et al., 2011), in the hydrological sciences (Han et
al., 2012; McLaughlin et al., 2002; McMillan et al., 2013; Park & Xu, 2013), and also in snow science (SNODAS; NOHRSC,
2004; Carroll et al., 2001). Data assimilation schemes in snow science rest on the notion that modeled variables like SWE can be
merged with an *in-situ* observed value at the same location and time using an objective function. This objective, or cost, function
quantifies the differences between the modeled state variable and the observed state (Reichle et al., 2002; Reichle et al., 2008;
McLaughlin, 2002). These methods can assimilate model state variables, like SWE, using a statistical method like a Kalman filter
or they can assimilate model fluxes like snowfall precipitation or snowmelt rates (Carroll et al., 2001; Clark et al., 2006; Magnussen





et al., 2014; Reichle et al., 2008). Other direct insertion assimilation schemes in snow science run the model twice, once without
the assimilated data, and a second time after the *in-situ* observations and correction factors are calculated in order to produce an
updated state variable (Liston et al., 2008; Malik et al., 2012; Helmert et al., 2018). Regardless of the method of assimilation, the
goal is the same: to produce a more accurate modeled state variable (snow depth or SWE) in space and time by using *in-situ*
observations to modify the process model output.

Snow depth measurements are a type of *in-situ* snowpack observation that can be made accurately and quickly by anyone with a
measuring device. As a consequence, the current study turns to citizen scientists for snow data collection. Citizen science is a
unique type of research in which scientists request input from the general public on data collection, data analysis, or data processing
(McKinley et al., 2017; Silvertown, 2009; Wiggins and Crowston, 2011). Through citizen science efforts, researchers access data
that are either highly decentralized or concentrated in space, as well as gather measurements frequently or randomly in time. The
primary advantage is that many people can accomplish data collection at spatial and temporal scales well beyond the capacity of a
single researcher or small group of scientists (Bonney et al., 2009; Cooper et al., 2007; Dickinson et al., 2010). Recent successful
citizen science-based research includes the CrowdHydrology project that monitors stage heights of streams and rivers (Fienen &
Lowry, 2012; Lowry & Fienen, 2013), and the CrowdWater project, which obtains multiple types of crowdsourced measurements
of hydrological variables using a publicly available app (Seibert et al., 2019; van Meerveld et al., 2017). Buytaert et al. (2014)
provides a comprehensive review of the recent challenges and motivations of citizen science in hydrology. This unique type of
data collected by citizen scientists has been used in many natural sciences, and snow hydrology represents a new opportunity for
citizen science-based research.

The present study explores the assimilation of a unique type of citizen science-based data in snow modeling: snow depth
measurements collected by citizen scientists traveling in snow covered landscapes worldwide. This new snow dataset and project
is called Community Snow Observations (CSO; communitysnowobs.org). The CSO campaign relies on backcountry recreationists
including skiers, snowboarders, snowmachiners, cross country skiers, snowshoers, and snow professionals, including avalanche
forecasters and snow scientists, who visit snowy environments for work and recreation to obtain snow depth measurements of the
snowpack (Hill et al., 2018; Yeeles, 2018). Other citizen science projects are underway in snow science, including research on the
relationship between vernal windows and snow depth (Contosta et al., 2017; Burakowski et al., 2018), snow depth verification of
satellite datasets in Canada using Twitter (Edmiston, 2012; Wiggins & Crowston, 2011), and the backyard precipitation
measurement campaign called Community Collaborative Rain, Hail, and Snow Network (Reges et al., 2016). The CSO project
adds to a growing body of research accomplished by citizen scientists in the natural sciences, and contributes to the connections
between physics-based, process modeling and *in-situ* observations in data assimilation and snow science.

The current study aims to answer two questions. First, can citizen scientists' snow depth measurements be incorporated into the
process model workflow in a way that improves model performance? This question is addressed by presenting an ensemble of
modeled snow depth and SWE distribution results with two types of outputs: (a) a set of model outputs without any snow depth
measurements assimilated and, (b) a set of model outputs with CSO snow depth measurements assimilated. To answer this first
question, we characterize the results using temporal and spatial datasets for validation. These datasets include time-series SWE
observations at a SNOTEL station in the study area and lidar- and photogrammetry-derived snow depth maps from 2017 and 2018.
We rely upon common metrics for characterizing the spatial distribution of modeled versus observed continuous environmental





variables to assess the value of the CSO modified outputs (Reimann et al., 2010). Secondly, how do the results vary with the
number of the CSO measurements assimilated? We address this question by randomly selecting and varying the quantity of CSO
measurements in the ensemble members. The potential of mobilizing a new type of *in-situ* snow dataset collected by snow
professionals and snow recreationists is significant because these participants often travel to remote mountainous environments
worldwide where *in-situ* snow observations are sparse.

**2 Study Area**
The study focuses on a 5,736 $km_2$ area of the eastern Chugach Mountains near Valdez, Alaska (Figure 1). This high-relief, glacier-
carved landscape ranges from sea-level in Port Valdez to rugged peaks exceeding 2200 m.a.s.l., and a mountain pass on the
Richardson Highway, named Thompson Pass (815 m.a.s.l). This region of the Chugach mountains receives extreme amounts of
snowfall, with Thompson Pass holding multiple snowfall records for the state of Alaska, including the 1-day total (1.57 m), 2-day
total (3.06 m), and weekly total (4.75 m; Shulski & Wendler, 2007). Like other places in the Chugach Mountains, snow densities
and snow depths in the region vary greatly across short distances (Wagner, 2012). There are deep, dense, and wet snowpacks found
in the maritime snow climates near the coast. The interior regions of the Chugach Mountains further from the coast contain
shallower, less-dense, and drier snow climates (Fieldwork 2018; Sturm et al., 1995; Sturm et al., 2010). These factors are important
because the Thompson Pass region and the Chugach mountains are frequently accessed by backcountry skiers and snowboarders,
backcountry snowmachiners, and multiple heli-skiing operations due to the exceptional access to steep terrain, and deep, mountain
snowpack (Carter et al., 2006; Hendrikx et al., 2016). Due to the popularity of the area for backcountry snowsports and the risk of
danger for avalanches affecting highway conditions, the Valdez Avalanche Center produces avalanche forecasts for many of the
slopes adjacent to the Richardson Highway in the Thompson Pass region. The choice of a study area within a mountainous region
visited regularly by snow recreationists and professionals is essential for the present study. For these reasons, the Thompson Pass
region of the Chugach Mountains in Alaska was selected for the initial phases of the CSO project.





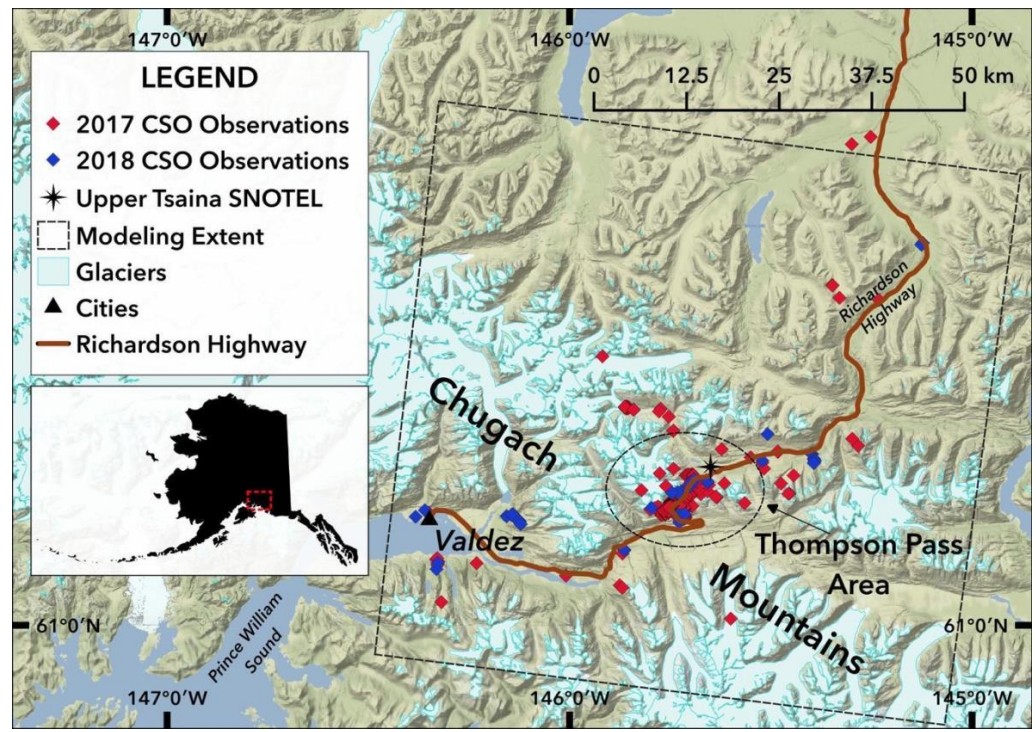

**Figure 1: Study Area Map.**
**The study area maps showing the Community Snow Observations (CSO) measurements, the modeling spatial extent, and the Thompson Pass region of the Chugach Mountains.**

## 3 Methods and Datasets

### 3.1 Model Dataflow

This study relies on a common research design in snow science that uses (1) *in-situ* snow observations, (2) physically-based process modelling, and (3) remote sensing of the snowpack to accomplish its primary objectives (Sturm et al., 2015). Figure 2 is a conceptual diagram of how the citizen scientists' snow depth measurements fit into the model chain for the present study. The modeling process begins with the weather forcing products and citizen scientists' snow depth observations as model inputs. Sub-models for meteorological variable distribution, snow depth to SWE estimation, and for the assimilation of snow measurements are employed before the final simulation occurs. The process model outputs are then validated by the RS datasets, the UTS station record, and the 2018 field measurements. Incorporating the citizen scientists' observations into the model chain is an attempt to modify the model outputs by *in-situ* snow depth observations.

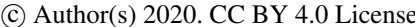



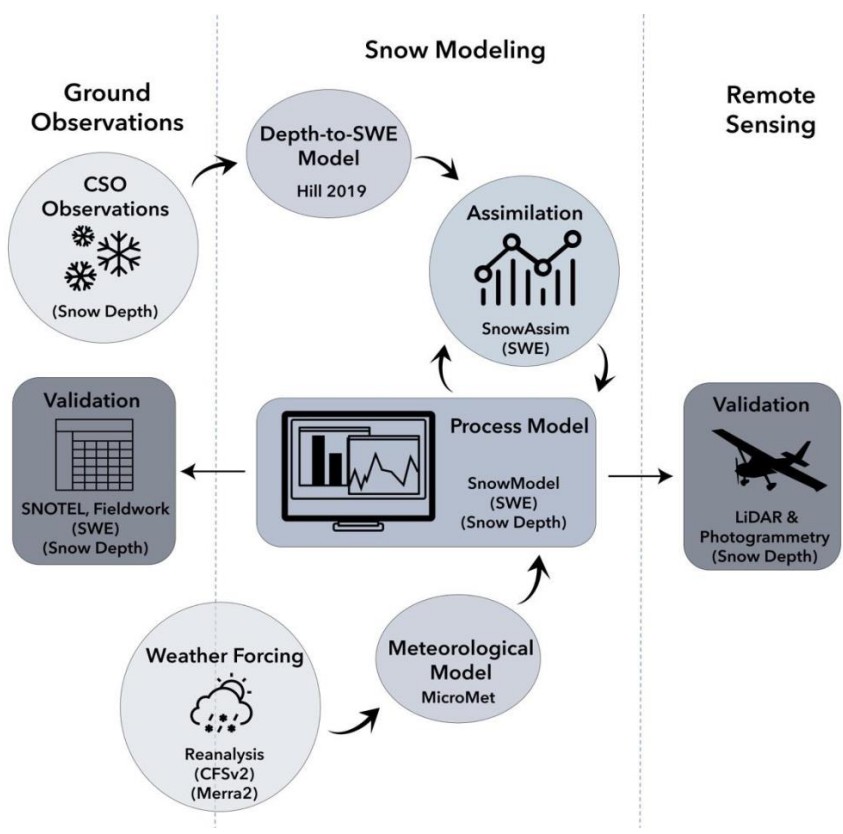

157

**Figure 2: Model Dataflow Diagram.**
**The model chain begins with the weather forcing product and the Community Snow Observations (CSO) datasets. The arrows indicate**
**dataflow through the series of sub-models to the process model output. The model output is then validated by the SNOTEL station**
**time-series, the 2018 fieldwork, and the remote sensing datasets.**

162

**3.2 Modeling Framework**

In this study we used a sequence of models to simulate SWE and snow depth distributions within the Thompson Pass study area

during WY2017 and WY2018. The sections below provide brief information about the models used in this study. For more details,

please refer to the source citations for each model.

167

**3.2.1 SnowModel**

SnowModel (Liston & Elder, 2006a) is a physically-based, spatially distributed process model for simulating the evolution of

snowpacks in snowy environments, and has been used for high-resolution and hemispheric-scale modeling worldwide (Beamer et

al., 2016; Beamer et al., 2017; Crumley et al., 2019; Liston & Heimstra, 2011; Mernild et al., 2017a-b). SnowModel is chosen for

the Chugach Mountains study area since it contains a data assimilation sub-model, SnowAssim, and a snow transportation sub-

model, SnowTran3d. Within SnowModel, various other sub-models solve the energy budget for the snowpack, generate runoff





quantities, etc. The present study focuses on the snow depth and SWE distribution outputs from SnowModel from simulations with
and without the data assimilation sub-model.

**3.2.2 MicroMet**
MicroMet (Liston & Elder, 2006b) is a meteorological distribution sub-model for weather station or reanalysis datasets that can be
paired with SnowModel in spatially explicit modeling applications. MicroMet uses the Barnes objective analysis scheme for
interpolating meteorological input variables to the gridded SnowModel domain for each model timestep (Barnes, 1964; Barnes,
1973). In the present study, instead of using weather station data, the model is forced with reanalysis data and MicroMet uses the
node locations as weather stations, accessing the reanalysis node surface level precipitation, wind speed and wind direction, relative
humidity, air temperature, and elevation variables for the spatial interpolation. MicroMet has been paired with reanalysis weather
products and SnowModel in many studies worldwide (Baha et al., 2018; Beamer et al., 2016; Liston & Heimstra, 2011; Mernild
et al., 2017a).

**3.2.3 SnowTran3d**
Wind redistribution of snow is an important factor for the spatial distribution of snow depths and SWE distributions for snow
modeling (Clark et al., 2011). Wind events build snow deposits in the gullies and the leeward side of bedrock features into drift
depths greater than 10 m at times within the Thompson Pass study area. These events also leave some portions of the landscape
completely scoured and void of snow based on fieldwork observations and the RS snow surveys from both years. SnowTran3d is
a sub-model within SnowModel that redistributes the snow laterally in the model grid according to the processes that govern snow
transportation: fetch, wind speed, wind direction, wind shear stress and the shear strength of the snowpack, saltation and turbulent
suspension of the snow, and sublimation (Liston et al., 2007). SnowTran3d is suitable for use as a sub-routine within SnowModel
when the model grid cell resolution is appropriate for the length scale of snow transportation processes to occur, for example,
primarily at model resolutions less than 100 m.

**3.2.4 SnowAssim**
To assimilate the CSO measurements, we used the sub-model SnowAssim developed in tandem with SnowModel (Liston and
Elder, 2008). For each water year (WY; defined as September 1st through August 31st) in the model time period, SnowModel
creates a full, preliminary simulation using the meteorological forcing dataset and no observational SWE data. Next, SnowAssim
compares the observed state SWE values at each location and time to the modelled state SWE values from the same grid locations
and time iterations. Note that CSO measurements are submitted as snow depths (m) and SnowAssim requires observational inputs
to be SWE depths (m), so a conversion from depth to SWE was necessary. The snow depth to SWE conversion method for the
current study will be discussed in the following section. SnowAssim aggregates all the assimilated observations by date and creates
a spatially varying correction surface that covers the entire model domain (Liston and Elder, 2008). These various correction
surfaces are applied by adjusting the model precipitation fluxes and snowmelt factors between SWE observation dates during a
second SnowModel simulation.






### 3.2.5 Snow Depth to Snow Water Equivalent Conversion

CSO participants take measurements of snow depth yet SnowAssim requires SWE observation inputs. A conversion from snow depth to SWE must be performed. A body of research exists on the best methods for converting point measurements from snow depth to SWE, using either bulk density estimations, snow climate classifications, statistical models, or atmospheric conditions and energy balance approaches (Sturm et al., 1995; Sturm et al., 2010; McCreight and Small 2014; Jonas et al., 2009; Pagano et al., 2009; Hill et al., 2019; Pistocchi, 2016). The Hill et al. (2019) model was chosen for two reasons. First, the data requirements are minimal for this model, requiring only location, day of water year (DOY) and readily-available climatological information based on input location. These minimal requirements align with the information available from CSO measurements. Second, it was found to outperform other bulk density methods such as Sturm et al. (2010) and Jonas et al. (2009) when tested against a wide variety of snow pillow and snow course datasets (Hill et al., 2019).

### 3.3 Model Input Datasets

### 3.3.1 Elevation and Land Cover

SnowModel requires a digital elevation model (DEM) and a land cover model as two of the three primary input datasets. The DEM is the National Elevation Dataset (NED) from the United State Geological Survey downloaded at 30 m resolution and then rescaled to 100 m spatial resolution (Gesch et al., 2002). The land cover model is the National Land Cover Database (NLCD) 2011 dataset at 30 m spatial resolution and then also resampled to 100 m resolution (Homer et al., 2011). The NLCD dataset is also reclassified to match the land cover input classes required by SnowModel. Initially, we test results from model simulations at two spatial resolutions, 30 m and 100 m, covering the model domain in the Thompson Pass region of the Chugach mountains. After calibrating the model, the results section only includes the 30m resolution.

### 3.3.2 Weather Forcing Datasets

Various weather reanalysis products have been used in remote portions of Alaska in previous studies (Beamer et al., 2016; Beamer et al., 2017; Crumley et al., 2019; Liston & Heimstra, 2011). In Alaska, each reanalysis product shows bias corresponding to meteorological variable, regional location, and season of the year (Lader et al., 2016; see their Figures 3 and 4). For this reason, the current study considered two weather reanalysis products that differ in their biases in temperature and precipitation in the Thompson Pass region during the winter and the summer seasons. We used the Climate Forecast System Reanalysis version 2 product (CFSv2) and the Modern-Era Retrospective Analysis for Research and Applications version 2 (MERRA2) product for the weather forcing inputs for SnowModel. The CFSv2 product from the National Centers for Environmental Prediction is an extension of the Climate Forecast System Reanalysis (CFSR) version 1 product that began in 1979, albeit at a lower spatial resolution (Saha et al., 2010). The CFSv2 data are available at a spatial resolution of 0.2 arc degrees, and a 6 hr temporal resolution (Saha et al., 2014). This CFSv2 dataset was downloaded using Google Earth Engine (GEE), a platform for accessing and analyzing scientific datasets with global coverage. The MERRA2 weather reanalysis product from NASA's Global Modeling and Assimilation office is the second meteorological forcing dataset tested in the present study (Gelaro et al., 2017). The MERRA2 data are available at a



spatial resolution of 0.667 degrees by 0.5 degrees, with a 3 hr temporal resolution beginning in 1979. MERRA2 replaces the older
version product with updated assimilation processes to include more weather datasets.

### 3.4 Snow Datasets

#### 3.4.1 Snow Telemetry Station Data

The study area contains two SNOTEL stations operated by NRCS. The first station is the Upper Tsaina SNOTEL (UTS) station
located at 534 m.a.s.l. on the NE side of Thompson Pass reporting the full standard set of sensor variables, including precipitation,
temperature, snow depth, and SWE. The second station is the Sugarloaf Mountain SNOTEL (SLS) station, located near the Valdez
Arm of the Prince William Sound at 168 m a.s.l. in the SW corner of the study area and records only precipitation, temperature,
and snow depth, but not SWE (Figure 1). Detailed information about the SNOTEL sensors and climate monitoring instruments
can be found at the SNOTEL website (https://www.wcc.nrcs.usda.gov/snow/) and Serreze et al. (1999). Direct links to the
SNOTEL websites for the UTS and SLS stations can also be found in the Data Availability section below.

#### 3.4.2 LiDAR and Photogrammetry Derived Data

The airborne photogrammetry survey was conducted on April 29, 2017 with a Nikon D800 36.2 megapixel camera and flown on
a fixed-wing aircraft above a portion of the Thompson Pass study area, see Figure 3 for location and extent. An onboard Trimble
Global Navigation Satellite System (GNSS) and a base-station were used for positional control. Post-processing was completed
with structure-from-motion software to create a digital surface model (DSM) of the photogrammetry-derived snow surface. The
airborne LiDAR survey was collected on April 7th and 8th, 2018, using a Riegl VUX1-LR laser scanner flown on a fixed-wing
aircraft. An onboard integrated inertial measurement unit (IMU) and GNSS, and a base-station were used to provide positional
control for the LiDAR-derived snow DSM. Both RS datasets were evaluated against a previously collected photogrammetry-
derived DSM from 2014 when no snow was present. An interpolation scheme was used to gap-fill some of the negative values in
the snow DSM due to vegetation cover effects.

#### 3.4.3 Chugach 2018 Fieldwork Data

Three weeks of fieldwork in the Thompson Pass region were conducted in March, April, and May of 2018. Snow depth and SWE
were measured throughout the study area with an avalanche probe and a Federal Snow Sampler. At each fieldwork measuring site,
a central SWE measurement was taken using the Federal Sampler. Avalanche probes were used in the surrounding 100 m2 to take
a series of 8 snow depth measurements extending 5 m in each direction from the central SWE measurement. The fieldwork
sampling protocol was designed to consider: (1) variability in snow depth in small areas less than 100 m2, (2) month-to-month
changes in snow depth and SWE, and (3) spatial gradients in snow density throughout the entire study area. A diagram of the
location of each observational site can be found in Figure 3. The 2018 fieldwork dataset was used for validation with two purposes
in mind. First, the 2018 fieldwork SWE measurements were used as a validation dataset for the 2018 SWE distribution results.
Secondly, since the data collected in the spring of 2018 contains measured snow depths and SWE at 70 observational sites (n =





560; 8 per site), we conducted an analysis of the sub-grid scale variability in snow depth found at each observational site and these
results are found in the discussion section.

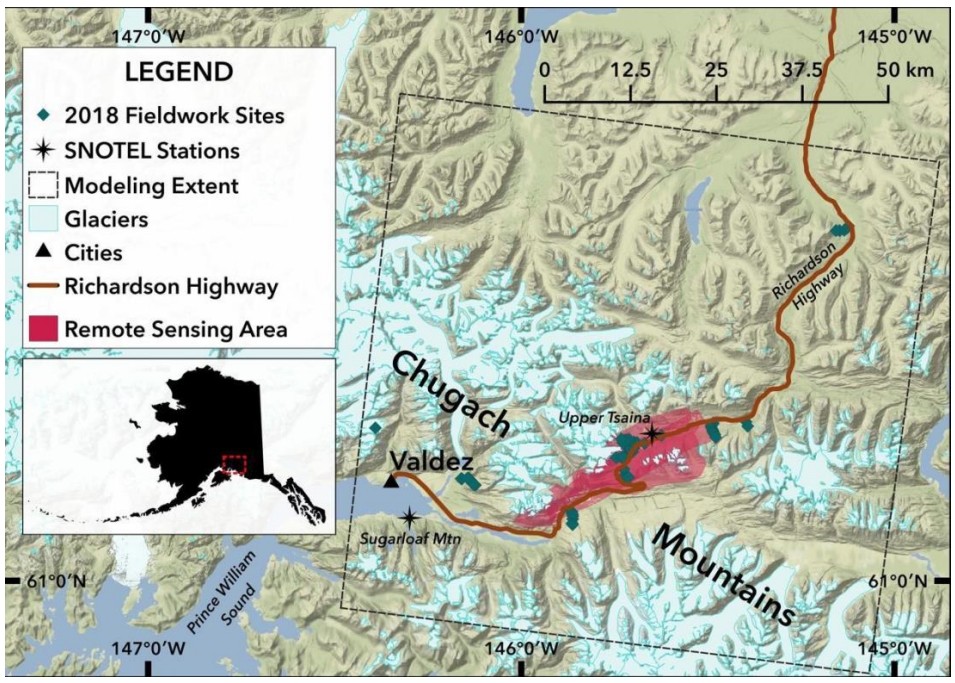


**Figure 3: Validation Datasets Map.**
**The 2018 fieldwork includes 72 sites with co-located snow water equivalent and snow depth measurements. The remote sensing**
**datasets from 2017 and 2018 are overlain on the map, along with the location of the Upper Tsaina SNOTEL station.**


### 3.4.4 Community Snow Observations Data

The CSO program collects snow depth data from citizen scientists in snowy environments worldwide. Full details including links
to smartphone apps and tutorials are found at http://communitysnowobs.org. Citizen scientists take several (2 to 4) snow depth
measurements within a small area (< 4 m2) using an avalanche probe or other depth measuring device (meterstick, etc.). These
measurements are then averaged by the participant and submitted using the app or program preferred by the participant. The
submitted data include the global positioning system (GPS) location in latitude and longitude, time and date, and snow depth
measurement (cm). The accuracy of the GPS system for each participants' mobile device determines the location error of the GPS,
with common errors for mobile phones ranging between +/- 4 to 7 m (Garnett & Stewart, 2015; Schaefer & Woodyer, 2015). Since
the model resolution is 30 m and 100 m, this level of horizontal error in GPS location is acceptable for the purposes of our research
questions. All collected data are made freely available on the CSO website for visualization and download (see Section 9 for Data
Availability). Thousands of measurements have been recorded by participants in CSO globally since it began in January 2017 with
initial measurement campaigns in Alaska and other frequently visited locations in mountain regions across North America (Figure
4). In the modeling domain of the current study, 442 CSO measurements were available for WY2017 and 104 CSO measurements


for WY2018. These measurements were concentrated in the Thompson Pass region of the study area (Figure 1) and range from 25
m to 1400 m in elevation.

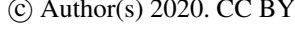


**Figure 4: CSO Participation in North America.**
**Participation in the Community Snow Observations (CSO) project in North America aggregated by the number of observations**
**recorded in each U.S. state or Canadian province between January 1st, 2017 and December 31st, 2019.**


### 4 Calibration

We performed model calibration using five years of the historical record of the UTS station from WY2012 through the end of
WY2016. The calibration was focused on adjustments to temperature lapse rates, precipitation lapse rates, wind adjustment factors,
and use of the SnowTran3d sub-model. We chose temperature lapse rates and precipitation lapse rates for calibration because
SnowModel is known to be limited by these factors when large elevational differences exist within the model domain (Liston and
Elder, 2006). We chose wind adjustment factors and the wind transportation sub-model for calibration because wind redistribution
of snow plays a significant role in the study area based on the 2018 fieldwork and the RS surveys from 2017 and 2018. Since the
SnowAssim sub-model requires a single layer snowpack, no adjustments were made to the snowpack layer structure. For each
weather reanalysis product a full calibration was performed for the 30m and 100m model resolutions, in the event that spatial
resolution plays a significant role in parameter selection. See Appendix A for the descriptions of the model parameters tested
during the calibration.

The daily SWE output from each calibration simulation is compared with the UTS observed SWE for the duration of the 5-year
calibration time period using root mean squared error (RMSE), the Nash Sutcliffe Efficiency (NSE), the Kling-Gupta Efficiency
(KGE), and mean bias error (Bias) to assess the calibration simulations. Table 1 lists the best 30m and 100m calibration simulations,
based on their time-series RMSE, NSE, KGE, and Bias scores. We acknowledge that measurement errors can occur with SNOTEL

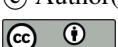



snow pillows and that these well known errors may affect the accuracy of the observational dataset (Johnson and Schaeffer, 2002;
Johnson, 2004).

326                                         **Table 1: Model Calibration Results.**
**The best calibration results are given for each set of simulations for water years 2012-2016, along with the root mean squared error**
**(RMSE), the Nash Sutcliffe Efficiency (NSE), the Kling-Gupta Efficiency (KGE), and the mean bias error (Bias).**

| Reanalysis Product & Resolution | Time Step | Number of Simulations | RMSE SWE (cm) | NSE | KGE | Bias SWE (+/- cm) |
|---|---|---|---|---|---|---|
| **MERRA2, 30m** | 3hrly | 45 | 24 | -0.29 | 0.08 | +16 |
| **MERRA2, 100m** | 3hrly | 45 | 26 | -0.10 | -0.10 | +19 |
| **CFSv2, 30m** | 6hrly | 45 | 22 | -0.15 | -0.01 | +17 |
| **CFSv2, 100m** | 6hrly | 45 | 22 | -0.15 | -0.01 | +17 |


Calibration results in Table 1 show that the 30m model grid resolution slightly outperforms the 100m model grid resolution in the
MERRA2-forced calibration simulations. However, the CFSv2-forced simulations show no difference between the model grid
resolutions. The CFSv2 product slightly outperforms the MERRA2 product in terms of SWE RMSE. Overall, the differences
between the top performing model grid resolution and reanalysis product are mixed and potentially negligible, varying by metric.
The NSE and KGE model performance metrics in the calibration simulations are lower than expected, due primarily to precipitation
inputs from the reanalysis products that were consistently higher than measured precipitation at the UTS station. Since SnowAssim
adjusts the precipitation fields during assimilation, these input deficiencies are acceptable for the purposes of this study. The
SnowModel default parameter values notably and consistently produce the top performing simulations, see Appendix B for details.
Due to each of these factors, the calibrated model for the remainder of the study uses the CFSv2 reanalysis product, the 30m model
grid resolution, and the SnowModel default parameter values.

One of the primary obstacles for process modeling is the use of accurate weather input data, and the related uncertainties with
weather inputs are a well-known complication in snow and hydrological modelling (Rivington et al., 2005; Schmucki et al., 2013;
Schlogl et al., 2016). Initial tests of modeled precipitation fields using Micromet versus the observed precipitation at the UTS
station revealed that both reanalysis products overestimated the amount of precipitation observed in the study area at the UTS
station. With these obstacles in mind, we designed an experiment to supplement the main findings of this research. For this
experiment we introduced a model precipitation adjustment factor similar to the method outlined in Mernild et al. (2006). We
applied this scalar value to the precipitation fields as a bias correction of the precipitation inputs. We tested 11 precipitation
adjustment factors ranging from 0.95 to 0.45 and applied them to the meteorological forcing inputs during the 5-year calibration
time period. For more details about the precipitation adjustment factor results, see Appendix C. This experiment, presented in
section 6.5, allows us test improvements to model performance when the precipitation inputs are bias corrected prior to model
assimilation of CSO measurements.





**5 Experimental Design**

With the model calibrated, we carried out a series of simulations in order to (1) quantify the improvement in model performance due to the assimilation of CSO measurements and to (2) understand the effects of the number of CSO data points selected for assimilation. Model simulations without using CSO measurements provide a baseline for comparison, referred to as the NoAssim case. Ensemble model simulations were also carried out with various numbers of CSO measurements assimilated, referred to as the CSO simulation case. An ensemble of 60 trials per year were carried out with n = 1, n = 2, n = 4, n = 8, n = 16, and n = 32, where n equals the number of CSO measurements assimilated per WY. In each instance (n value), 10 realizations of the numerical experiment were carried out.

The timeframe of the assimilating CSO measurements was restricted to the peak SWE period or later. According to the UTS station, peak SWE in the study area generally occurs mid- to late-April and consequently the earliest assimilation date was set to April 15th. The CSO measurements were aggregated by week because initial simulations suggested that daily increments were not producing realistic results by SnowAssim. Additionally, CSO participation in the Thompson Pass region during the early accumulation season was infrequent in WY2018 and non-existent in WY2017. Since peak SWE is important for mountain hydrology and ecology, with many snow studies using it as an indicator metric, the time restrictions are acceptable for the research questions addressed in this study (Bohr and Aguado, 2001; Trujillo et al., 2012; Kapnick and Hall, 2012; Mote et al., 2018; Wrzesien et al., 2017).

**6 Results**

The following results reflect the three types of available validation datasets: 1) time-series SWE results at the UTS station, 2) spatial snow depth distributions from the RS datasets, and 3) point-based snow depth and SWE measurements from the 2018 fieldwork.

**6.1 Temporal Results Using the Upper Tsaina SNOTEL Station**

The temporal results compare the UTS station SWE time-series to the ensemble member SWE time-series during WY2017 and WY2018. Figure 5 displays the temporal cycle of snowpack accumulation and ablation, and the timing of peak SWE. At the UTS station in the study area, the average WY day of peak SWE is 228, or April 15th. Before this day, the snowpack is generally increasing in SWE and afterwards the snowpack generally enters the ablation period with a reduction in SWE. This temporal cycle can be observed in Figure 5 by following the color gradient. The highest performing (Best) CSO simulation (Figure 5b,e) corrects the slope of the snowpack accumulation and ablation phases when contrasted with the NoAssim accumulation and ablation phases and slopes (Figure 5a,d). These time-series results, in terms of model performance metrics and the snowpack temporal cycle, exhibit SnowAssim's ability to incorporate CSO measurements and improve modeled SWE outputs at the UTS station location throughout the entire snow season.





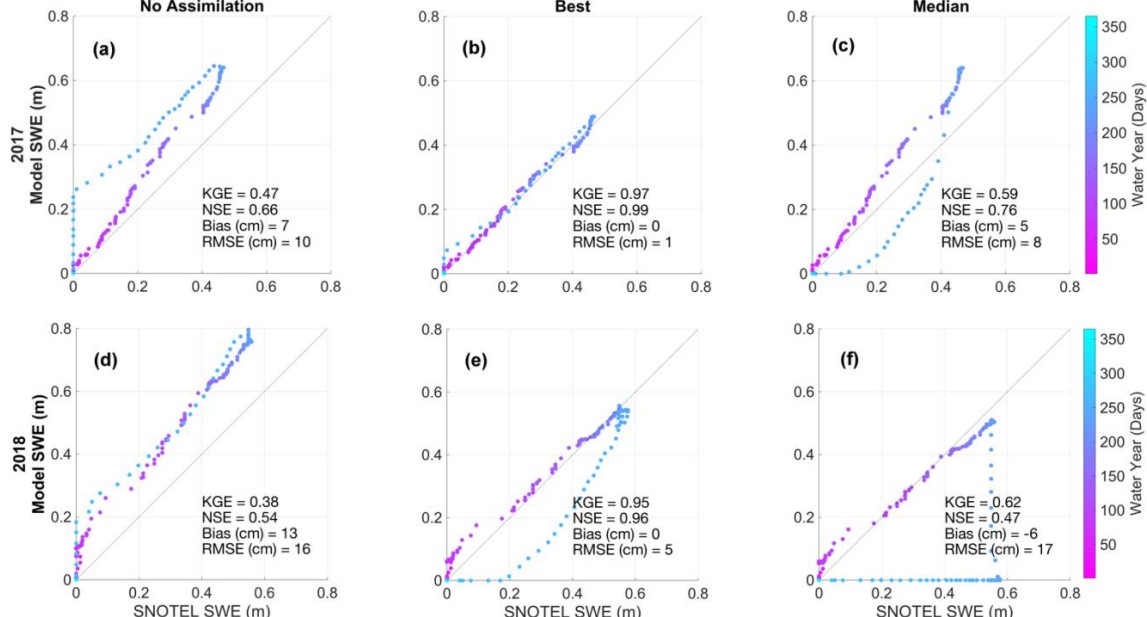

**Figure 5: Time Series at Upper Tsaina SNOTEL Station.**
**The Upper Tsaina SNOTEL snow water equivalent (SWE) observations versus the modeled SWE for the no assimilation case (a,d), the**
**Best CSO simulation (b,e), and the Median CSO simulation (c,f). The timeseries color gradient corresponds to the day of the water**
**year.**

Figure 5 summarizes the temporal results for the Best and median performing (Median) CSO simulations, including the NoAssim
case. Each ensemble member is evaluated by their KGE, NSE, RMSE, and Bias scores. For results presented in this section, the
KGE score is used to rank the ensemble simulations. A full accounting of each ensemble member and their time-series ranking can
be found in Appendix D. Modeled SWE depths for the NoAssim case are consistently higher than the UTS station SWE
observations for both WYs (Figure 5a,d). The modeled SWE depths for the Best CSO simulation outperform the NoAssim case
throughout the entirety of the time-series and represent an improvement in model performance scores according to all of the time-
series metrics (Figure 5b,e). The modeled SWE depths for the Median CSO simulation for WY2017 outperform the NoAssim case
by all metrics, and the WY2018 Median CSO results are mixed. The ensemble simulation KGE scores outperform the NoAssim
KGE scores among 70% of the WY2017 ensemble members, and among 67% of the WY2018 ensemble members. Any number
of CSO measurements assimilated show improvements in model performance, a key finding in the time-series results.

**6.2 Spatial Results Using the Remote Sensing Datasets**
The ensemble results are summarized in Figure 6 using the Kolmogorov-Smirnov statistic (KS; Massey 1951). The KS statistic
quantifies the difference between a reference dataset of a continuous variable and a sample dataset of the same variable. The KS
statistic represents the maximum distance between the empirical cumulative distribution function (ECDF) of the reference and
sample datasets, with KS scores ranging from zero to one, with zero representing perfect dataset agreement (Reimann et al., 2010).
In the KS analysis, the reference dataset is the RS derived snow depth distribution and the sample datasets are each of the ensemble





snow depth distributions, including the NoAssim case. Figure 6 shows that in WY2017 the CSO simulations are an improvement
from the 2017 NoAssim case among 62% of the ensemble members, and in WY2018 among 78% of the ensemble members. Note
that only the KS values that fall below the NoAssim line represent an improvement in model performance during the CSO
simulations. The spatial results reveal that improvements in model performance are not dependent upon the number of CSO
measurements that are assimilated in WY2018. However, WY2017 has a smaller range in KS values as the number of assimilated
measurements increases, with more CSO simulations outperforming the NoAssim case. These results also vary according to model
performance metric and by WY, with no clear pattern emerging from the number of measurements assimilated.

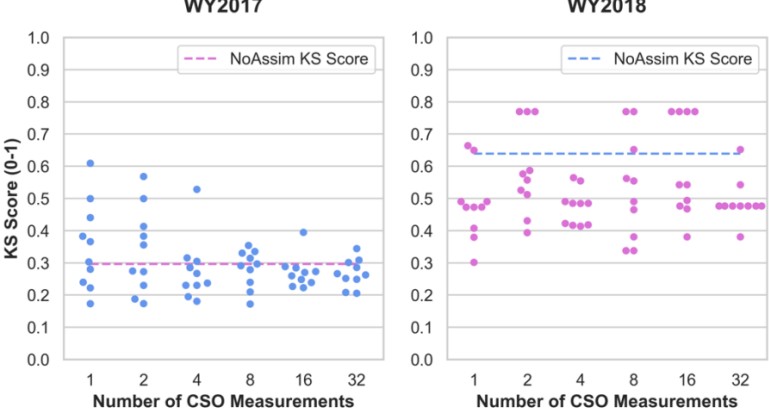

419                                    **Figure 6: Swarmplots of Kolomogorov-Smirnov Scores.**
**The ensemble simulations are ranked by Kolmogorov-Smirnov (KS) score per year and plotted according to the number of**
421                          **measurements assimilated, including the no assimilation (NoAssim) case.**


The snow depth distribution maps in Figure 7 display the RS datasets (a,b), the results from the highest performing CSO simulation
(c,d), and the NoAssim case for each WY (e,f). Refer to Figure 2 for the RS dataset location within the study area. We present the
Best CSO simulation as the focus of Section 6.2 ranked according to KS score ranking (Figure 6). A full accounting of each
ensemble member and their spatial distribution ranking can be found in Appendix E. In the RS datasets, there is more variation
and heterogeneity in snow depth across short distances (Figure 7a-b). This spatial diversity is evident even after the RS dataset has
been aggregated to correspond to the model resolution at 30 m, as depicted in Figure 7. The NoAssim case and Best CSO simulation
show less spatial diversity, and the NoAssim case broadly overstimates snow depth when compared to the Best CSO simulation
for both WYs. The visualization of the snow depth distributions in Figure 7 illustrate the challenges of accurately representing the
process scale through physics-based modeling at low resolutions (Blöschl 1999), and some of these challenges will be examined
further in the discussion section.




**Figure 7: Snow Depth Distribution Maps.**
(a,b) The remote sensing (RS) datasets from 2017 and 2018. (c,d) The best CSO simulation results corresponding to the RS dataset spatial extent. (e,f) The no assimilation results corresponding to the RS dataset spatial extent. The total model area that corresponds to the RS dataset in 2017 is 104 km$_2$ and 149 km$_2$ in 2018.






Figure 8 presents  histograms and empirical cumulative distribution functions (ECDFs) for the RS datasets, the NoAssim case, and
the Best CSO simulation. In WY2017 (Figure 8a), when the NoAssim case overestimates snow depths, the Best CSO simulation
ECDF shifts left, towards the RS dataset ECDF. To a greater degree, in WY2018 (Figure 8c) when the NoAssim case more broadly
overestimates the snow depths, the Best CSO simulation ECDF shifts further left, towards the RS dataset ECDF. The shifts in the
EDCFs are evident in the histograms and the median value of each dataset is indicated with a dashed line (Figure 8b,d). The same
shifts are evident in the snow depth distribution maps (Figure 7c,d,e,f). Even though the shifts in ECDFs and histograms are in the
correct direction in the Best CSO simulations, SnowAssim is not adjusting the distribution of snow depth values, which can be
seen in the multimodal shape of the histograms.

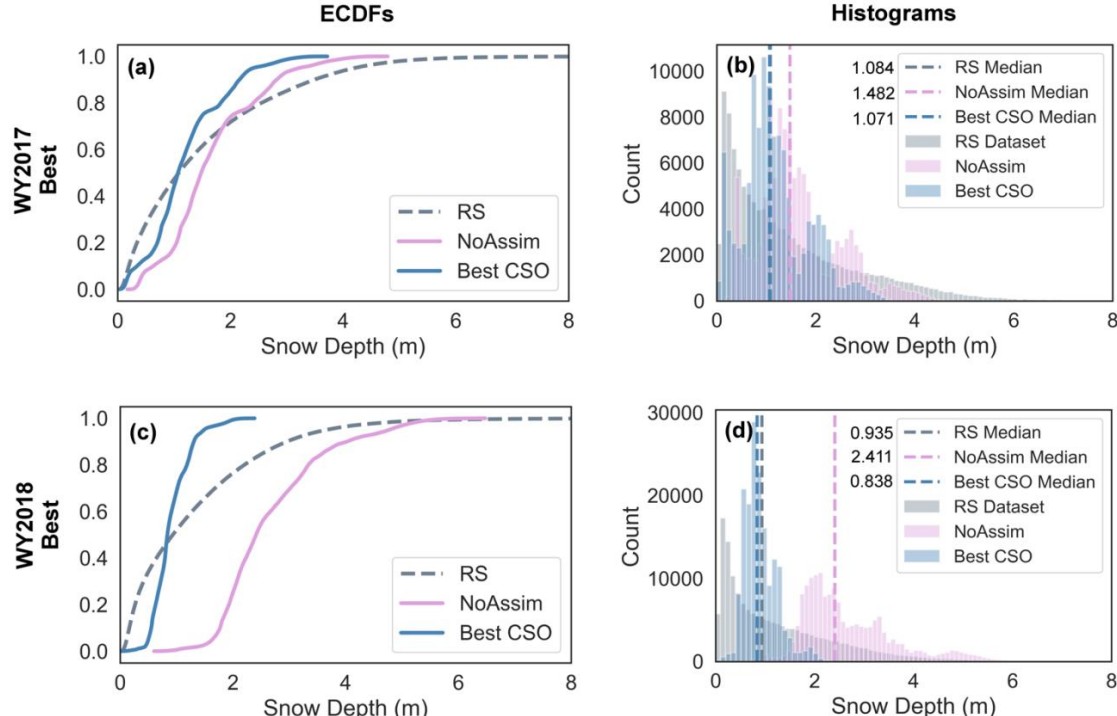


**Figure 8: Histogram and Distribution Plots.**
**The empirical cumulative distribution functions (ECDFs) and histograms from the best CSO simulation, the no assimilation case, and**
**the remote sensing (RS) datasets during WY2017 (a,b) and WY2018 (c,d).**

The multimodal distribution of snow depths in the modeled results can be explained by their relationship to the elevation of the
surrounding terrain. The input DEM and the snow depth distributions were compared on a grid-cell-to-grid-cell basis using a two-
dimensional histogram (2DH). Figure 9 is a series of 2DHs that display snow depth (x axes) versus the input DEM (y axes) in the
RS area from both years. Darker colors indicate a higher frequency of snow depth and elevation values corresponding to each
dataset. The 2DHs show a proportional relationship between the modeled snow depths (Figure 9 a,b,e,f) and the input DEM values.
As elevation increases, snow depth also increases linearly in the modeled results. Still, the range of snow depths from Best CSO

simulation shifts towards the RS dataset in both years, but the elevation relationship remains largely intact. The RS snow depths
are less dependent on elevation, with snow depth values between 0 and 1 appearing at all elevations between 0 and 1250m. The
2DH analysis supports the findings from the snow depth distribution maps where the variability of snow depth observed in the RS
dataset is not replicated in the NoAssim case or the Best CSO simulation (Figure 7).

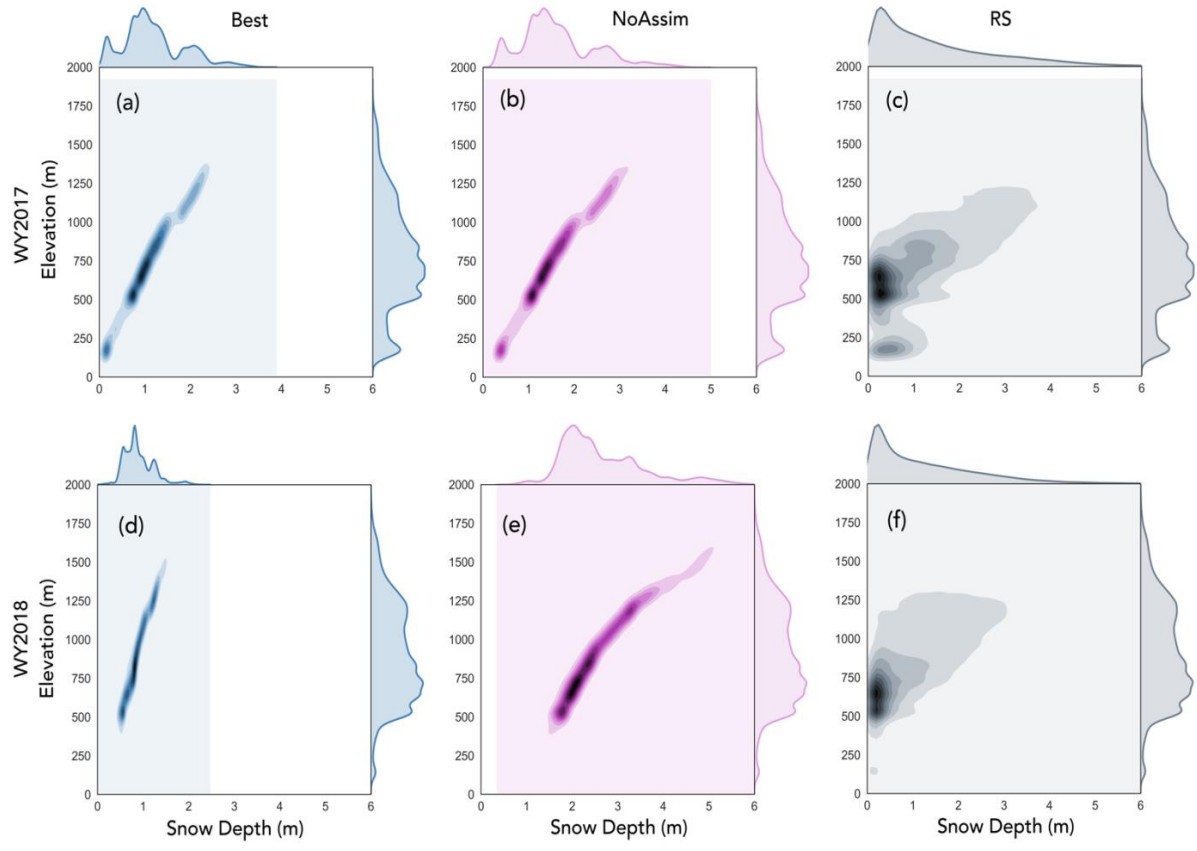

**Figure 9: Two-dimensional Histograms.**
**The remote sensing (RS) dataset vs. the (a) water year (WY) 2017 no assimilation case, (b) WY2018 no assimilation case, (c) WY2017**
**best CSO simulation, and (d) WY2018 best CSO simulation.**

**6.3 Fieldwork 2018 Results**

To validate the WY2018 SWE distributions from the NoAssim case and the Best CSO simulation we used ground-truth data from
our field campaign in April 2018. The locations of the 70 SWE and snow depth measurement sites from 2018 are depicted in
Figure 3. Figure 10 shows the co-located SWE depth measurements (y axes) versus the snow depth measurements (x axes) from
each site aggregated by month. The bars in Figure 10 represent the variability in snow depth within the surrounding $100m_2$ of the
SWE measurement, including the average, minimum, and maximum of 8 snow depth measurements at each site. Table 3 shows
the results at the SWE measurement sites, comparing the NoAssim case versus the Best CSO simulation using RMSE, bias, and
mean absolute error (MAE) metrics for evaluation. Since each measurement site corresponds to a single CSO snow depth





submission, we separated those measurement sites used in the assimilation scheme from the validation set when creating Table 3.
The Best CSO simulation outperforms the NoAssim case according to all metrics in all months. The 2018 fieldwork results from
April show that the Best CSO simulation has a bias of +3 cm, while the NoAssim case is +97 cm. The April 2018 fieldwork results
agree with the histogram and ECDF analysis that displayed broad overestimation of SWE in the NoAssim case in WY2018 (Figure
7b; Figure 8d).

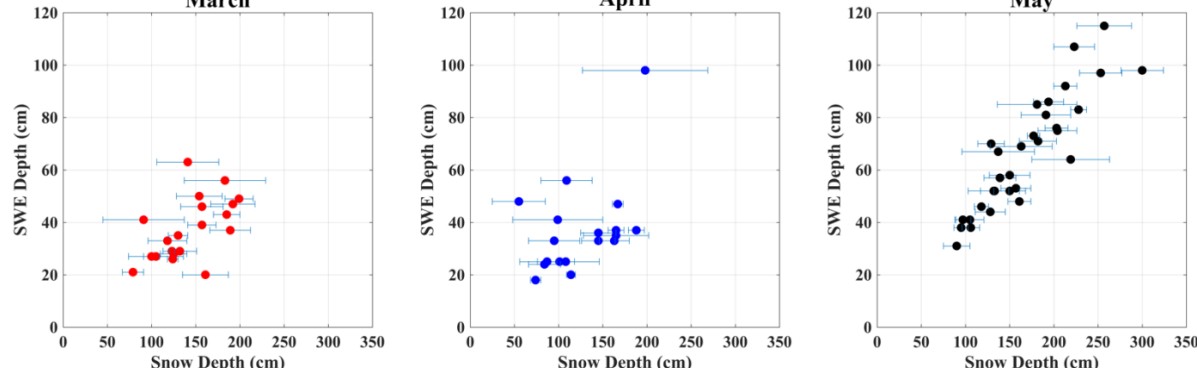

**Figure 10: Fieldwork 2018 Measurements by Month**
**The 70 *in-situ* snow water equivalent (SWE) measurements (y axes) from 2018 are plotted by month along with their co-located snow**
**depth measurements (x axes). The bars show the minimum, maximum, and average of each fieldwork site where 8 snow depth**
**measurements were obtained in a 100 m$_2$ area.**
**Table 3: Fieldwork 2018 Results**
**The 70 SWE measurements from the 2018 fieldwork compared to the Best CSO simulation and the no assimilation (NoAssim) case**
**using the three model performance metrics: root mean squared error (RMSE), mean bias error (Bias), and mean absolute error**
**(MAE).**

|  | Bias SWE (cm) | | RMSE SWE (cm) | | MAE SWE (cm) | |
|---|---|---|---|---|---|---|
|  | **Best CSO** | **NoAssim** | **Best CSO** | **NoAssim** | **Best CSO** | **NoAssim** |
| **All** | -11 | 86 | 28 | 100 | 22 | 86 |
| **March** | -3 | 77 | 15 | 95 | 13 | 77 |
| **April** | 3 | 97 | 21 | 114 | 16 | 97 |
| **May** | -25 | 84 | 37 | 95 | 31 | 84 |


**6.4 Spatially Averaged Snow Water Equivalent Results**

Another way to quantify the ability of CSO measurements to constrain SnowModel output is to investigate the modeled SWE
averaged over a large area. Table 4 contains the spatially averaged SWE estimations from the RS survey area in WY2018, and
includes the RS dataset, the Best CSO simulation, and the NoAssim case. We focus on WY2018 because the fieldwork
measurements include estimated bulk density values at each measurement site. These bulk density estimations were measured
during April 2018 and were partitioned from the larger dataset and spatially averaged over the RS region only (n=22). The
fieldwork estimated bulk density value was then applied to the spatially averaged RS snow depth. For the Best CSO simulation
and the NoAssim case, the spatially averaged snow depth, SWE, and snow density values were taken directly from the model





results. The SWE estimation results in Table 4 demonstrate that SnowAssim can constrain the SWE output over a large region
based on a few, randomly chosen CSO measurements. Importantly, the accuracy of the total modeled water volume from the RS
region in 2018 improves when CSO measurements are included, a key finding that has implications for water resource management
decisions in snowy, data-limited, mountain environments.

**Table 4: Spatially Averaged Variables in the RS Region**
**The spatially averaged results were calculated using the RS region in WY2018, the RS dataset, and the modeled results. The spatially**
**averaged SWE depth for the RS survey was estimated using the average density measured during April 2018 fieldwork.**

| Dataset | Spatially Averaged Snow Depth (cm) | Spatially Averaged Density (kg/m³) | Spatially Averaged SWE Depth (cm) | Total RS Region Water Volume (km³) |
|---|---|---|---|---|
| **RS Survey 2018** | 130 (RS survey) | 331 (fieldwork) | 43 (estimated) | 0.06 (estimated) |
| **Best CSO Simulation 2018** | 130 (modeled) | 400 (modeled) | 52 (modeled) | 0.08 (modeled) |
| **NoAssim 2018** | 267 (modeled) | 430 (modeled) | 115 (modeled) | 0.17 (modeled) |



### 6.5 Precipitation Adjustment Experiment

The experimental design of the present study was developed for remote locations where a long-term precipitation dataset was not
available to bias correct the precipitation inputs. However, since a long-term precipitation dataset may be available in other
locations, we decided to test the results with a precipitation experiment. In this experiment we applied a scalar to the CFSv2
precipitation fields for bias correction and all other model parameters and input datasets were held constant. The experiment results
show that some of the CSO ensemble simulations still outperformed the NoAssim case with the precipitation adjustment, both
spatially and temporally. For example, the spatial results show that 43% percent of the ensemble runs in WY2017 and 20% of the
ensemble runs in WY2018 outperformed the NoAssim case when the precipitation was bias corrected, according to their KS score
(Figure 11). Similarly, the temporal results show that 42% of the ensemble runs in WY2017 and 58% of the ensemble runs in
WY2018 outperformed the NoAssim case when the precipitation was bias corrected, according to their KGE score. The ECDF
and histogram analysis from the precipitation adjustment factor experiment also show model improvements when there was broad
underestimation of snow depths in the NoAssim case in WY2017 and broad overestimation in WY2018. These results demonstrate
that using CSO measurements for assimilation can improve model performance when the available weather forcing dataset has
known biases (no precipitation adjustment factor case) but when those biases have been decreased (precipitation adjustment factor
case) the improvements become less clear, they vary from year to year, and are less consistent between spatial and temporal results.





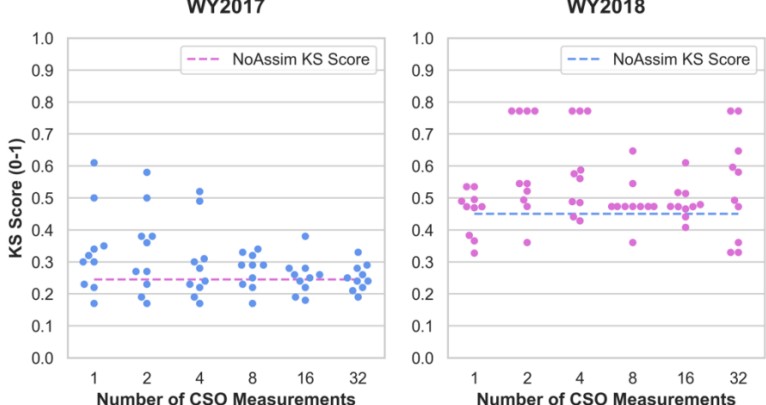



**Figure 11: Swarmplots of Kolmogorov-Smirnov Scores with Precipitation Adjustment Factor.**
**The ensemble simulations are ranked by Kolmogorov-Smirnov (KS) score per water year (WY) and plotted according to the number**
**of CSO measurements assimilated, including the no assimilation (NoAssim) case.**


### 7 Discussion

An important consideration in the results of the present study involves ranking the CSO ensemble members by various spatial and
temporal metrics. The time series results (Section 6.1), the spatially distributed results (Section 6.2), and the spatially averaged
results (Section 6.4) did not have the same ranking order for the CSO ensemble members. For example, the Best CSO simulation
in WY2017 from the time-series analysis was an ensemble member with two CSO measurements assimilated according to the
KGE metric. The time-series results represent a single point in the domain, the UTS station. By contrast, the Best CSO simulation
in WY2017 from the spatial distribution analysis was an ensemble member with eight CSO measurements assimilated using the
KS score. The spatially distributed results represent the entire RS survey area. The improvements in model performance are
determined by the type of validation dataset available and the metric used to quantify those improvements. In other words, one
size does not fit all when it comes to quantifying improvements to model performance using CSO measurements.

The variability of snow depth and SWE in mountain catchments and the spatial patterning of snowpack conditions in complex
terrain is a well-known challenge in snow modeling and snow remote sensing research (Anderton et al., 2004; Lopez-Moreno et
al., 2013; Luce et al., 1998; Molotch et al., 2005; Rice and Bales, 2010; Sturm et al., 2010b). The RS results reveal that variability
in snow depth across short distances is largely a function of wind redistribution and drifting and not primarily a function of elevation
(Figure 9c,f; Figure 7a,b). Thompson Pass is a notoriously windy location, and the RS dataset shows complex drifting patterns
throughout the surveyed area (Figure 7a,b). The wind inputs from the reanalysis product used in Micromet and SnowTran3d may
not be adequate for the steepness and ruggedness of the terrain. Although wind scaling factors were tested in the calibration, the
only suitable calibration dataset was the SNOTEL site. SNOTEL stations are often situated in locations where the effects of wind
redistribution of the snowpack are dampened and SNOTEL station data are often not representative of the spatial variability of the
surrounding areas (Dressler et al., 2006; Molotch and Bales, 2005). The inability to of SnowTran3d to resolve the wind





redistribution of the snowpack more accurately, the course wind field inputs from the reanalysis products, and the use of a single
SNOTEL station for calibration, together represent a model and input data limitation of the current study.

The ensemble results highlight a deeper question in snow hydrology and process modeling in general, regarding the sub-grid scale
variability of the modeled state variable within a single model grid cell. The scale of the *in-situ* observations (measured with an
avalanche probe) and the scale of the model resolution (30 m grid) versus the scale of the physical process being modeled (true
patterns and true variance in space and time) can create scale effects that need to be accounted for (Blöschl et al., 1999). In this
way, the 2018 fieldwork has a significant role to play in our understanding of the sub-grid scale variability in snow depth
distributions. CSO participants average a few point measurements over a 1-4 $m_2$ area. The model resolution is 30 m, or 900 $m_2$ per
grid model grid cell. If participants move slightly one direction or another, their averaged and submitted measurements would
likely be different, but their measurements would potentially lie within the same 30 m model grid cell. This difference, in turn,
would modify the SWE depth inputs for SnowAssim. To better characterize the sub-grid scale variability of snow depth we
investigate the 8 avalanche probe depths taken over 100 $m_2$ at each of the 70 observation sites during the 2018 fieldwork (see also
Figure 11). From these data, a picture of the sub-grid scale variability emerges. The largest range in snow depth values at a single
100 $m_2$ observation site is 2.11 m and the smallest range in snow depth values at a single site is 0.09 m. The highest standard
deviation (sd) found at a single observation site is 0.71 m and the lowest sd is 0.04 m. This shows that a significant amount of
variation, and therefore uncertainty, is being added to the model chain simply by the sub-grid scale variability of snow depth
distributions within a single model grid cell, distributions that the model will not be able to resolve at the 30 m or 100 m resolution.
Sub-grid scale variability is a well known problem in snow science and represents a limitation of the improvements that can be
made by assimilating CSO measurements (Elder et al., 1993, Blöschl et al., 1999; Liston et al., 2008; Schmucki et al., 2013).

One of the limitations of the present study is that the physical and temporal characteristics of the CSO measurements like aspect,
elevation, and early-season measurements were not fully tested. Initial simulations demonstrated that SnowAssim performs best
when the assimilated measurements were located close in time to the validation dataset. This factor influenced our choice to focus
on the late-season time period of CSO measurements since the RS surveys were conducted in the late-season. Additionally, since
the majority of the CSO measurements for both WYs occurred between March 15th and May 15th, future research should be in a
location where CSO measurements are obtained frequently throughout the accumulation season. A research project with many
measurements throughout the accumulation period may provide more insights into the temporal aspects of assimilation of CSO
measurements. We decided not to subset the CSO measurements by geophysical characteristics like aspect, elevation, and land
cover type because these require additional analysis that is outside of the scope of the current study. Understanding the effects of
temporal and spatial restrictions of CSO measurements on model performance will likely be an area of future research.
Additionally, it may be necessary to test other process models and alternate assimilation schemes in the future to improve the
spatial distribution of model results and determine if CSO measurements can be used in other modeling contexts.

**7 Conclusions**
In this study we use a new snow dataset collected by participants in the Community Snow Observations (CSO) project in coastal
Alaska to improve snow depth and snow water equivalence (SWE) outputs from a snow process model. Ensemble simulations



were carried out during the 2017 and 2018 snow seasons to investigate the effects of incorporating citizen science measurements
into the model chain using an assimilation scheme. Time series SNOTEL station records, remotely sensed photogrammetry and
light detection and ranging surveys, and fieldwork observations are used to validate the modeled snow depth and snow water
equivalent distributions. Any number of CSO measurements assimilated improves model performance, from 1 to 32. Our results
demonstrate that using CSO measurements for assimilation can improve model performance when the available weather forcing
dataset has known biases and also when those biases have been decreased by using a precipitation adjustment factor. The
improvements in model performance from CSO measurements occur in 62% to 78% of the ensemble simulations both spatially
and temporally, and in cases when the model broadly overestimates or underestimates snow depth and SWE. Model estimations
of total water volume from a sub-region of the study area also demonstrate improvements in accuracy after CSO measurements
have been assimilated. This study has implications for water resource management and snow modeling in locations where *in-situ*
snow information is limited but snow enthusiasts often visit, since even small numbers of assimilated CSO measurements can
improve the snow model outputs.
**8 Appendices**
**Appendix A: Model calibration parameters and their descriptions.**

| Parameter | # of Options | Format | Description |
|---|---|---|---|
| Temperature Lapse Rate | 3 sets | Monthly | PRISM Climatologies; Local Weather Station Data; SnowModel Default |
| Precipitation Lapse Rate | 5 sets | Monthly | Monthly Coefficients of ¼, ½, ¾, 1(SnowModel Default), PRISM Climatologies |
| Wind Adjustment Factor | 3 | Coefficient | Coefficients of 1(SnowModel Default),2,3 |
| SnowTran3d | 2 | On/Off | |



**Appendix B: Top performing parameter configurations from the calibration simulations.**

| Rank | Temperature Lapse Rate | Precipitation Scaling Factor | Wind Adjustment Factor | SnoTran on/off |
|---|---|---|---|---|
| Tied for first | Default | Default | Default | On |
| Tied for first | Local Weather Station | Default | Default | On |
| Tied for first | PRISM Climatologies | Default | Default | On |



**Appendix C: Precipitation Adjustment Factor Results.**
**The best precipitation adjustment factors are shown, along with the root mean squared error (RMSE), the Nash Sutcliffe Efficiency**
**(NSE), the Kling-Gupta Efficiency (KGE), and the mean bias error (Bias).**

| Reanalysis, Resolution | Time Period (WY) | Time Step | Number of Simulations | Precipitation Adjustment Factor | RMSE Precipitation (mm) | NSE | KGE | Bias Precipitation (+/- mm) |
|---|---|---|---|---|---|---|---|---|
| **MERRA2, 30m** | 2012-2016 | 3hrly | 11 | 0.55 | 7.5 | 0.07 | 0.20 | 0.0 |
| **MERRA2, 100m** | 2012-2016 | 3hrly | 11 | 0.55 | 7.5 | 0.07 | 0.20 | 0.0 |
| **CFSv2, 30m** | 2012-2016 | 6hrly | 11 | 0.60 | 6.7 | 0.27 | 0.35 | -0.1 |
| **CFSv2, 100m** | 2012-2016 | 6hrly | 11 | 0.60 | 6.7 | 0.27 | 0.35 | -0.1 |







**Appendix D: Ranked Temporal Results.**
**Ensemble results from ranked by Kling-Gupta efficiency (KGE) score for water year (WY) 2017 (a) and WY2018 (b). Also included**
**are the Nash Sutcliffe Efficiency (NSE) and the mean bias error (Bias) values.**
**(a) WY2017**

| Rank | Number of CSO Measurements | Iteration | KGE | NSE | Bias (cm) |
|---|---|---|---|---|---|
| 1 | 2 | 2 | 0.97 | 0.99 | 0 |
| 2 | 1 | 8 | 0.97 | 0.99 | 0 |
| 3 | 4 | 1 | 0.94 | 0.93 | 0 |
| 4 | 2 | 6 | 0.93 | 0.92 | 0 |
| 5 | 8 | 9 | 0.93 | 0.89 | -1 |
| 6 | 16 | 8 | 0.90 | 0.84 | -1 |
| 7 | 32 | 3 | 0.88 | 0.96 | -1 |
| 8 | 4 | 4 | 0.88 | 0.91 | -2 |
| 9 | 1 | 10 | 0.80 | 0.95 | -3 |
| 10 | 4 | 3 | 0.80 | 0.89 | 2 |
| 11 | 16 | 2 | 0.78 | 0.82 | -3 |
| 12 | 8 | 1 | 0.77 | 0.81 | 2 |
| 13 | 32 | 8 | 0.77 | 0.79 | -3 |
| 14 | 2 | 8 | 0.77 | 0.93 | -3 |
| 15 | 16 | 7 | 0.76 | 0.93 | -3 |
| 16 | 16 | 1 | 0.75 | 0.87 | -3 |
| 17 | 4 | 6 | 0.74 | 0.92 | -3 |
| 18 | 1 | 6 | 0.71 | 0.89 | 4 |
| 19 | 16 | 3 | 0.67 | 0.88 | -4 |
| 20 | 32 | 4 | 0.66 | 0.79 | -5 |
| 21 | 32 | 5 | 0.65 | 0.78 | -5 |
| 22 | 32 | 1 | 0.65 | 0.78 | -5 |
| 23 | 32 | 7 | 0.64 | 0.80 | -5 |
| 24 | 2 | 3 | 0.63 | 0.80 | 4 |
| 25 | 4 | 9 | 0.62 | 0.83 | -5 |
| 26 | 16 | 9 | 0.62 | 0.82 | -5 |
| 27 | 2 | 10 | 0.61 | 0.82 | -5 |
| 28 | 16 | 4 | 0.60 | 0.75 | -5 |
| 29 | 32 | 6 | 0.59 | 0.82 | -5 |
| 30 | 8 | 8 | 0.59 | 0.76 | 5 |
| 31 | 32 | 2 | 0.57 | 0.78 | 6 |
| 32 | 16 | 5 | 0.56 | 0.73 | -6 |
| 33 | 4 | 8 | 0.56 | 0.73 | -6 |
| 34 | 8 | 10 | 0.55 | 0.72 | -6 |
| 35 | 8 | 7 | 0.54 | 0.73 | -6 |
| 36 | 16 | 6 | 0.54 | 0.70 | -6 |
| 37 | 1 | 3 | 0.54 | 0.74 | 6 |
| 38 | 8 | 2 | 0.52 | 0.68 | -6 |
| 39 | 8 | 4 | 0.52 | 0.71 | -6 |
| 40 | 1 | 2 | 0.51 | 0.72 | -6 |
| 41 | 4 | 10 | 0.50 | 0.67 | -7 |
| 42 | 32 | 10 | 0.49 | 0.66 | -7 |
| 43 | 4 | 7 | 0.46 | 0.63 | -7 |
| NoAssim | NoAssim | NoAssim | 0.47 | 0.66 | 7 |
| 44 | 8 | 3 | 0.43 | 0.66 | -7 |
| 45 | 32 | 9 | 0.41 | 0.63 | -8 |
| 46 | 8 | 5 | 0.39 | 0.54 | -8 |
| 47 | 2 | 1 | 0.36 | 0.53 | -8 |
| 48 | 8 | 6 | 0.34 | 0.49 | -9 |
| 49 | 1 | 4 | 0.33 | 0.49 | -9 |
| 50 | 1 | 7 | 0.29 | 0.42 | -9 |





| 51 | 2 | 4 | 0.28 | 0.41 | -9 |
|---|---|---|---|---|---|
| 52 | 16 | 10 | 0.26 | 0.37 | -10 |
| 53 | 2 | 5 | 0.22 | 0.32 | -10 |
| 54 | 1 | 5 | 0.17 | 0.23 | -11 |
| 55 | 1 | 9 | 0.08 | 0.05 | -12 |
| 56 | 2 | 7 | 0.08 | 0.05 | -12 |
| 57 | 4 | 2 | 0.06 | 0.02 | -12 |
| 58 | 4 | 5 | 0.03 | -0.03 | -12 |
| 59 | 2 | 9 | -0.02 | -0.13 | -13 |
| 60 | 1 | 1 | -0.07 | -0.24 | -14 |



**(b) WY2018**

| Rank | Number of CSO Measurements | Iteration | KGE | NSE | Bias (m) |
|---|---|---|---|---|---|
| 1 | 2 | 7 | 0.95 | 0.96 | 0 |
| 2 | 8 | 9 | 0.91 | 0.90 | 2 |
| 3 | 8 | 5 | 0.90 | 0.89 | 2 |
| 4 | 2 | 9 | 0.88 | 0.91 | 2 |
| 5 | 2 | 4 | 0.87 | 0.93 | -2 |
| 6 | 4 | 7 | 0.87 | 0.97 | 3 |
| 7 | 4 | 8 | 0.84 | 0.97 | -2 |
| 8 | 1 | 5 | 0.84 | 0.95 | -2 |
| 9 | 1 | 6 | 0.84 | 0.95 | -2 |
| 10 | 4 | 10 | 0.82 | 0.95 | 4 |
| 11 | 2 | 2 | 0.77 | 0.92 | 5 |
| 12 | 4 | 9 | 0.77 | 0.88 | -4 |
| 13 | 16 | 9 | 0.76 | 0.85 | -4 |
| 14 | 16 | 5 | 0.76 | 0.53 | -2 |
| 15 | 16 | 4 | 0.76 | 0.53 | -2 |
| 16 | 4 | 6 | 0.75 | 0.84 | -4 |
| 17 | 32 | 10 | 0.74 | 0.49 | -2 |
| 18 | 4 | 5 | 0.71 | 0.72 | -5 |
| 19 | 2 | 6 | 0.71 | 0.89 | 6 |
| 20 | 1 | 8 | 0.71 | 0.83 | -5 |
| 21 | 1 | 1 | 0.71 | 0.83 | -5 |
| 22 | 1 | 9 | 0.71 | 0.83 | -5 |
| 23 | 8 | 7 | 0.69 | 0.80 | -6 |
| 24 | 16 | 8 | 0.68 | 0.58 | -6 |
| 25 | 16 | 2 | 0.65 | 0.77 | -6 |
| 26 | 32 | 2 | 0.65 | 0.53 | -6 |
| 27 | 32 | 5 | 0.64 | 0.50 | -6 |
| 28 | 32 | 8 | 0.64 | 0.49 | -6 |
| 29 | 32 | 7 | 0.62 | 0.47 | -6 |
| 30 | 32 | 9 | 0.62 | 0.47 | -6 |
| 31 | 32 | 4 | 0.62 | 0.46 | -6 |
| 32 | 32 | 1 | 0.62 | 0.46 | -6 |
| 33 | 8 | 10 | 0.57 | 0.42 | -7 |
| 34 | 4 | 1 | 0.53 | 0.65 | -9 |
| 35 | 2 | 1 | 0.52 | 0.65 | -9 |
| 36 | 32 | 3 | 0.49 | 0.18 | 6 |
| 37 | 4 | 4 | 0.48 | 0.60 | -10 |
| 38 | 4 | 2 | 0.47 | 0.60 | -10 |
| 39 | 4 | 3 | 0.45 | 0.57 | -10 |
| 40 | 8 | 6 | 0.43 | 0.52 | 11 |
| 41 | 2 | 3 | 0.38 | 0.46 | -11 |
| 42 | 1 | 7 | 0.33 | 0.38 | -12 |
| 43 | 8 | 4 | 0.30 | 0.29 | -13 |
| 44 | 1 | 2 | 0.30 | 0.36 | 15 |
| 45 | 16 | 1 | 0.24 | 0.14 | -14 |
| 46 | 32 | 6 | 0.24 | 0.13 | -14 |
| 47 | 1 | 4 | 0.23 | 0.29 | 16 |





| | | | | | |
|---|---|---|---|---|---|
| 48 | 1 | 10 | 0.07 | -0.09 | -17 |
| 49 | 8 | 8 | 0.01 | -0.21 | -18 |
| 50 | 8 | 3 | 0.00 | -0.24 | -18 |
| 51 | 1 | 3 | -0.07 | -0.37 | -20 |
| 52 | 16 | 3 | -0.15 | -1.18 | 18 |
| 53 | 16 | 7 | -0.16 | -1.15 | 18 |
| 54 | 16 | 6 | -0.16 | -1.15 | 18 |
| 55 | 8 | 1 | -0.16 | -1.14 | 18 |
| 56 | 16 | 10 | -0.16 | -1.13 | 19 |
| 57 | 2 | 8 | -0.23 | -1.05 | 21 |
| 58 | 8 | 2 | -0.28 | -1.07 | 23 |
| 59 | 2 | 5 | -0.37 | -1.18 | 27 |
| 60 | 2 | 10 | -0.58 | -2.00 | 32 |



**Appendix E: Ranked Spatial Results.**
**Spatial distribution ensemble results ranked by Kolmogorov-Smirnov (KS) score for water year (WY) 2017 (a) and WY2018 (b). Also**
**included are the root mean squared error (RMSE) and the median values.**
**(a) WY2017 Results**

| Rank | Number of CSO Measurements | Iteration | KS Score (0 - 1) | RMSE (m) | Median (m) | Mean (m) |
|---|---|---|---|---|---|---|
| 1 | 8 | 9 | 0.17 | 1.171 | 1.071 | 1.198 |
| 2 | 1 | 8 | 0.17 | 1.173 | 1.066 | 1.192 |
| 3 | 2 | 2 | 0.17 | 1.173 | 1.064 | 1.190 |
| 4 | 4 | 1 | 0.18 | 1.164 | 1.096 | 1.225 |
| 5 | 2 | 6 | 0.19 | 1.159 | 1.116 | 1.248 |
| 6 | 4 | 4 | 0.19 | 1.202 | 0.983 | 1.100 |
| 7 | 32 | 2 | 0.21 | 1.149 | 1.156 | 1.393 |
| 8 | 32 | 3 | 0.21 | 1.222 | 0.931 | 1.044 |
| 9 | 8 | 8 | 0.21 | 1.148 | 1.166 | 1.402 |
| 10 | 1 | 10 | 0.22 | 1.243 | 0.888 | 0.995 |
| 11 | 16 | 8 | 0.22 | 1.287 | 0.693 | 0.883 |
| 12 | 16 | 1 | 0.23 | 1.251 | 0.872 | 0.978 |
| 13 | 2 | 8 | 0.23 | 1.256 | 0.861 | 0.966 |
| 14 | 4 | 2 | 0.23 | 1.135 | 1.250 | 1.396 |
| 15 | 4 | 3 | 0.23 | 1.135 | 1.250 | 1.396 |
| 16 | 4 | 6 | 0.24 | 1.267 | 0.840 | 0.942 |
| 17 | 16 | 7 | 0.24 | 1.270 | 0.834 | 0.936 |
| 18 | 8 | 1 | 0.24 | 1.133 | 1.281 | 1.430 |
| 19 | 1 | 6 | 0.24 | 1.133 | 1.281 | 1.430 |
| 20 | 16 | 2 | 0.25 | 1.321 | 0.651 | 0.814 |
| 21 | 32 | 4 | 0.25 | 1.293 | 0.801 | 0.891 |
| 22 | 32 | 5 | 0.25 | 1.293 | 0.794 | 0.892 |
| 23 | 16 | 3 | 0.26 | 1.306 | 0.770 | 0.866 |
| 24 | 32 | 1 | 0.26 | 1.310 | 0.761 | 0.855 |
| 25 | 32 | 7 | 0.27 | 1.316 | 0.754 | 0.847 |
| 26 | 4 | 9 | 0.27 | 1.320 | 0.749 | 0.843 |
| 27 | 16 | 4 | 0.27 | 1.324 | 0.738 | 0.832 |
| 28 | 2 | 10 | 0.27 | 1.328 | 0.731 | 0.825 |
| 29 | 16 | 9 | 0.27 | 1.328 | 0.730 | 0.824 |
| 30 | 2 | 3 | 0.27 | 1.135 | 1.406 | 1.567 |
| 31 | 8 | 10 | 0.28 | 1.344 | 0.715 | 0.804 |
| 32 | 1 | 3 | 0.28 | 1.137 | 1.426 | 1.589 |
| 33 | 16 | 5 | 0.28 | 1.349 | 0.696 | 0.788 |
| 34 | 4 | 8 | 0.29 | 1.350 | 0.694 | 0.786 |
| 35 | 32 | 6 | 0.29 | 1.351 | 0.692 | 0.784 |
| 36 | 16 | 6 | 0.29 | 1.355 | 0.685 | 0.777 |
| 37 | 8 | 7 | 0.29 | 1.360 | 0.678 | 0.769 |
| NoAssim | NoAssim | NoAssim | 0.30 | 1.145 | 1.482 | 1.651 |
| 38 | 8 | 2 | 0.30 | 1.370 | 0.663 | 0.753 |


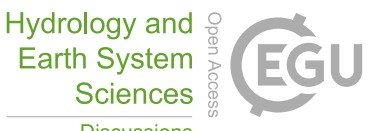

| Rank | Number of CSO Measurements | Iteration | KS Score (0 - 1) | RMSE (m) | Median (m) | Mean (m) |
|---|---|---|---|---|---|---|
| 39 | 32 | 10 | 0.30 | 1.384 | 0.649 | 0.731 |
| 40 | 1 | 2 | 0.30 | 1.381 | 0.644 | 0.734 |
| 41 | 4 | 10 | 0.30 | 1.384 | 0.639 | 0.729 |
| 42 | 32 | 8 | 0.31 | 1.404 | 0.461 | 0.667 |
| 43 | 8 | 4 | 0.31 | 1.400 | 0.614 | 0.703 |
| 44 | 4 | 7 | 0.32 | 1.402 | 0.612 | 0.701 |
| 45 | 8 | 3 | 0.33 | 1.426 | 0.573 | 0.662 |
| 46 | 8 | 5 | 0.34 | 1.438 | 0.565 | 0.649 |
| 47 | 32 | 9 | 0.34 | 1.448 | 0.546 | 0.630 |
| 48 | 8 | 6 | 0.35 | 1.469 | 0.521 | 0.603 |
| 49 | 2 | 1 | 0.36 | 1.468 | 0.514 | 0.600 |
| 50 | 1 | 4 | 0.37 | 1.484 | 0.490 | 0.576 |
| 51 | 1 | 7 | 0.38 | 1.510 | 0.453 | 0.539 |
| 52 | 2 | 4 | 0.38 | 1.510 | 0.453 | 0.539 |
| 53 | 16 | 10 | 0.39 | 1.529 | 0.426 | 0.512 |
| 54 | 2 | 5 | 0.41 | 1.559 | 0.385 | 0.472 |
| 55 | 1 | 5 | 0.44 | 1.601 | 0.330 | 0.418 |
| 56 | 1 | 9 | 0.50 | 1.684 | 0.223 | 0.314 |
| 57 | 2 | 7 | 0.50 | 1.684 | 0.223 | 0.314 |
| 58 | 4 | 5 | 0.53 | 1.724 | 0.175 | 0.268 |
| 59 | 2 | 9 | 0.57 | 1.770 | 0.119 | 0.217 |
| 60 | 1 | 1 | 0.61 | 1.812 | 0.067 | 0.173 |






**(b) WY2018 Results**

| Rank | Number of CSO Measurements | Iteration | KS Score (0 - 1) | RMSE (m) | Median (m) | Mean (m) |
|---|---|---|---|---|---|---|
| 1 | 1 | 10 | 0.30 | 1.210 | 0.838 | 0.905 |
| 2 | 8 | 3 | 0.34 | 1.246 | 0.756 | 0.810 |
| 3 | 8 | 8 | 0.34 | 1.246 | 0.756 | 0.810 |
| 4 | 1 | 7 | 0.38 | 1.146 | 1.124 | 1.238 |
| 5 | 16 | 1 | 0.38 | 1.150 | 1.127 | 1.237 |
| 6 | 32 | 6 | 0.38 | 1.150 | 1.127 | 1.237 |
| 7 | 8 | 4 | 0.38 | 1.150 | 1.127 | 1.237 |
| 8 | 2 | 3 | 0.39 | 1.146 | 1.182 | 1.304 |
| 9 | 1 | 3 | 0.41 | 1.319 | 0.621 | 0.655 |
| 10 | 4 | 3 | 0.41 | 1.153 | 1.261 | 1.392 |
| 11 | 4 | 1 | 0.42 | 1.147 | 1.292 | 1.437 |
| 12 | 4 | 2 | 0.42 | 1.155 | 1.279 | 1.413 |
| 13 | 4 | 4 | 0.42 | 1.165 | 1.305 | 1.435 |
| 14 | 2 | 1 | 0.43 | 1.166 | 1.335 | 1.474 |
| 15 | 8 | 7 | 0.46 | 1.205 | 1.487 | 1.651 |
| 16 | 16 | 2 | 0.47 | 1.261 | 1.568 | 1.708 |
| 17 | 1 | 1 | 0.47 | 1.221 | 1.521 | 1.684 |
| 18 | 1 | 9 | 0.47 | 1.221 | 1.521 | 1.684 |
| 19 | 1 | 8 | 0.47 | 1.221 | 1.523 | 1.686 |
| 20 | 16 | 8 | 0.48 | 1.233 | 1.553 | 1.746 |
| 21 | 32 | 1 | 0.48 | 1.233 | 1.553 | 1.746 |
| 22 | 32 | 2 | 0.48 | 1.233 | 1.553 | 1.746 |
| 23 | 32 | 4 | 0.48 | 1.233 | 1.553 | 1.746 |
| 24 | 32 | 5 | 0.48 | 1.233 | 1.553 | 1.746 |
| 25 | 32 | 7 | 0.48 | 1.233 | 1.553 | 1.746 |
| 26 | 32 | 8 | 0.48 | 1.233 | 1.553 | 1.746 |
| 27 | 32 | 9 | 0.48 | 1.233 | 1.553 | 1.746 |
| 28 | 4 | 9 | 0.48 | 1.244 | 1.577 | 1.753 |
| 29 | 4 | 5 | 0.48 | 1.248 | 1.580 | 1.748 |
| 30 | 4 | 6 | 0.48 | 1.248 | 1.580 | 1.748 |
| 31 | 1 | 5 | 0.49 | 1.259 | 1.607 | 1.780 |



| 32 | 1 | 6 | 0.49 | 1.259 | 1.607 | 1.780 |
| 33 | 4 | 8 | 0.49 | 1.259 | 1.607 | 1.780 |
| 34 | 8 | 10 | 0.49 | 1.259 | 1.607 | 1.780 |
| 35 | 16 | 9 | 0.49 | 1.281 | 1.628 | 1.801 |
| 36 | 2 | 4 | 0.51 | 1.318 | 1.714 | 1.893 |
| 37 | 2 | 7 | 0.53 | 1.353 | 1.777 | 1.968 |
| 38 | 16 | 4 | 0.54 | 1.401 | 1.848 | 2.068 |
| 39 | 16 | 5 | 0.54 | 1.401 | 1.848 | 2.068 |
| 40 | 32 | 10 | 0.54 | 1.401 | 1.848 | 2.068 |
| 41 | 8 | 9 | 0.55 | 1.453 | 1.922 | 2.131 |
| 42 | 4 | 7 | 0.55 | 1.454 | 1.928 | 2.132 |
| 43 | 2 | 9 | 0.56 | 1.461 | 1.939 | 2.148 |
| 44 | 8 | 5 | 0.56 | 1.500 | 1.977 | 2.189 |
| 45 | 4 | 10 | 0.56 | 1.493 | 1.980 | 2.191 |
| 46 | 2 | 2 | 0.58 | 1.540 | 2.043 | 2.263 |
| 47 | 2 | 6 | 0.59 | 1.606 | 2.128 | 2.350 |
| **NoAssim** | **NoAssim** | **NoAssim** | 0.64 | 1.861 | 2.411 | 2.678 |
| 48 | 1 | 2 | 0.65 | 1.894 | 2.436 | 2.721 |
| 49 | 32 | 3 | 0.65 | 1.928 | 2.466 | 2.764 |
| 50 | 8 | 6 | 0.65 | 1.928 | 2.466 | 2.764 |
| 51 | 1 | 4 | 0.66 | 2.009 | 2.567 | 2.852 |
| 52 | 16 | 10 | 0.77 | 2.932 | 3.466 | 3.839 |
| 53 | 16 | 3 | 0.77 | 2.932 | 3.466 | 3.839 |
| 54 | 16 | 6 | 0.77 | 2.932 | 3.466 | 3.839 |
| 55 | 16 | 7 | 0.77 | 2.932 | 3.466 | 3.839 |
| 56 | 2 | 10 | 0.77 | 2.932 | 3.466 | 3.839 |
| 57 | 2 | 5 | 0.77 | 2.932 | 3.466 | 3.839 |
| 58 | 2 | 8 | 0.77 | 2.932 | 3.466 | 3.839 |
| 59 | 8 | 1 | 0.77 | 2.932 | 3.466 | 3.839 |
| 60 | 8 | 2 | 0.77 | 2.932 | 3.466 | 3.839 |


## 9 Code and Data Availability

The datasets used in this study can be found at the following locations.

1. Community Snow Observations website and snow depth data download at http://app.communitysnowobs.org/
(last accessed 30 April 2020).

2. The snow depth to snow water equivalence calculator (Hill et al., 2019) can be downloaded via Github at
https://github.com/communitysnowobs/snowdensity (last accessed: 30 April 2020).

3. Snow Telemetry data for the Upper Tsaina River station near Valdez, Alaska is available at the Natural Resources
Conservation Service website: https://wcc.sc.egov.usda.gov/nwcc/site?sitenum=1055 (last accessed: 30 April 2020).

4. Climate Forecast System Reanalysis version 2 (CFSv2) data (Saha et al., 2011) is available for download at
https://rda.ucar.edu/datasets/ds094.0/#!description.

5. The CFSv2 data was accessed using Google Earth Engine at https://developers.google.com/earth-
engine/datasets/catalog/NOAA_CFSV2_FOR6H (last accessed: 30 April 2020). A javascript version of the Earth Engine



code written for this project is available at https://github.com/snowmodel-tools/preprocess_javascript (last accessed: 30
April 2020).

6.  To convert the CFSv2 data downloaded from Google Earth Engine to the necessary input file for MicroMet we
wrote Matlab scripts that can be downloaded via Github at https://github.com/snowmodel-tools/preprocess_matlab (last
accessed: 30 April 2020).

7.  The MERRA2 weather reanalysis product from NASA's Global Modeling and Assimilation office (Gelaro et
al., 2017) can be downloaded at https://gmao.gsfc.nasa.gov/reanalysis/MERRA-2/data_access/ (last accessed: 30 April
661  2020).


8.  The National Elevation Dataset is (Gesch et al., 2002) available for download at
https://catalog.data.gov/dataset/usgs-national-elevation-dataset-ned (last accessed: 30 April 2020).

9.  The National Land Cover Database 2011 dataset (Homer et al., 2011) is available for download at the Multi-
Resolution Land Characteristics Consortium at https://www.mrlc.gov/data?f%5B0%5D=category%3Aland%20cover
(last accessed: 30 April 2020).
**10 Author Contributions**
Ryan Crumley, David Hill, Gabriel Wolken, Katreen Wikstrom Jones, and Anthony Arendt designed the research questions and
decided on the methods. Ryan Crumley, Gabriel Wolken, Katreen Wikstrom Jones, and David Hill conducted fieldwork in the
study area, including snowpack sampling and remote sensing surveys. Ryan Crumley and Dave Hill oversaw the analysis of the
manuscript. Anthony Arendt designed and maintained the CSO website and snow dataset with contributions from all authors.
Community Snow Observation Participants and all authors contributed snow depth measurements. Ryan Crumley prepared the
manuscript with contributions from all authors during editing and review process.
**11 Competing Interests**
The authors declare that they have no conflicts of interest.
**12 Acknowledgements**
This research has been supported by NASA (grant no. NNX17AG67A) and CUAHSI (Pathfinder Fellowship grant). Arendt was
partially supported by the Washington Research Foundation, and by a Data Science Environments project award to the University
of Washington eScience Institute from the Gordon and Betty Moore and the Alfred P. Sloan Foundations.

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
