# Peer review of "Assimilation of citizen science data in snowpack modeling using a new snow dataset: Community Snow Observations"

_Hydrology and Earth System Sciences, 2020_

## Referee Comment (RC1) · Anonymous Referee #1 · 8 Dec 2020

The article is interesting and innovative. The use of data measured by the community is a contribution to the simulation of snow distribution and a way of bringing the community closer to snow science and hydrology.

The scientific quality of the article is good; however, the article could improve the analysis on some topics described below.

First, despite mentioning that the distribution of snow by the wind is important, the article do not present results or analysis in this regard.

Snowmodel allows you to export the results of wind redistribution. Showing these results would be a contribution to the analysis and discussion. Also, a comparison with

a simulation without wind redistribution (Windtrans off) would be a way to measure the improvement of using this tool.

Secondly, the assimilation in Snowmodel is highly depending of swe point location, in addition to timing. It is important to consider in the analysis where the data used are located. And if they agree in time and place with the validation dataset.

if the SWE data used for assimilation are located close to the validation point. Logically the result will be very similar to the validation point measure since the model correct the precipitation or fusion to obtain a value close to the given one. For this reason, it is important to know how close is the CSO data the field work data. If these two data are very close in time and location it does not make sense to use the field work data for validation.

Finally, the article should include a comparison between the data used: RS, CSO and field work data. The objective is to check if the data are consistent with each other and if they are very similar in time and location. Also, the article should include include a comparison between the densities estimated to convert the CSO data to snow water equivalent and the densities measured in the field work.

Some specific comments:

0) Figure 1 and 3 should be next to each other or join them to be able to compare the distribution of the data used for assimilation and validation

1) Point 3.2.5 Snow depth to snow water equivalent conversion. Add the uncertainty in the snow density estimation

2) Point 6 why the Sugarloaf Mountain station is not used to validate the results?

3) Point 6.2 The location or spatial distribution of CSO measurement used for the assimilation is as important as the number and should be and it should be analyzed here or elsewhere.

4) Line 459 (Figure 9 a,b,d,e)

---

## Referee Comment (RC2) · Anonymous Referee #2 · 14 Dec 2020

The manuscript describes use of a new citizen science dataset (snow depth) to guide simulations of SWE via data assimilation (DA). The motivation is to include observations gathered from locations in the landscape that might not be monitored otherwise. The study is focused on a maritime snow climate of Alaska. The model used (Snow-Model) has a long history and is well established. A range of different observations (Snotel, field surveys with depth and SWE, and remotely-sensed snow depth) are used to gauge performance of the DA system, compared to simulations that do not include the depth observations.

The results presented are interesting and there appears to be considerable potential

for use of the citizen science depth observations. However, major revisions could help make the manuscript more useful. The following issues should be addressed:

First, a more complete description of the DA approach is needed. In the introduction, a more detailed comparison of the approach used relative to other snow DA efforts should be provided – beyond what is currently included in the introduction (e.g., L80). How does the approach used compare to other methods, including direct insertion (e.g. Hedrick et al., 2018), particle-batch smoother (Margulis et al. 2019), particle filter (Smyth et al. 2019) and possibly EnKF. The methods section provides only a limited description of how the model is adjusted for mismatch with observations ("SnowAssim aggregates all the assimilated observations by date and creates a spatially varying correction surface that covers the entire model domain (Liston and Elder, 2008). These various correction surfaces are applied by adjusting the model precipitation fluxes and snowmelt factors between SWE observation dates during a second SnowModel simulation"). The 'adjustments' to the model are central to the effort, so the method should be described more completely in the manuscript. The results (or discussion) do not include any documentation of the 'adjustments' to the model, yet one of the benefits of DA is that the merging of data and models is one way to more completely understand the entire system (e.g., see Magnusson et al., 2014 and 2017 and their retrieved precipitation correction factor).

Second, uncertainty of the observations and validation data should be described and incorporated into the analysis. One of the benefits of DA is that the magnitude of uncertainty can be explicitly included in the analysis (e.g., Magnusson et al., 2014 and 2017). It appears that uncertainty of the assimilated observations is not included in the analysis – is this the case? If not, why not? The spatial representativeness of the depth measurements is mentioned in the discussion. One component of uncertainty is related to the conversion from depth to SWE, using the density estimation described in Hill (2019). In the region analyzed, SWE estimates based on density from Hill (2019) have an RMSE of 0.2-0.25 (normalized to snow season precipitation). Is this

considered in the DA approach? Uncertainty (or biases) of the validation data is not described, thus it is implied that the data are 'perfect'. What is the error or uncertainty associated with the federal sampler data?

Third, something seems strange about the calibration and validation methods and results. Are the NSE values in Table 1 correct? If the best simulation has NSE < 0, this would suggest that the calibration is not working very well. Additional details are required. Is calibration for the entire year? The entire snow year? Why not at peak SWE? Results in Fig 5 also seem strange. Fig 5e: how can this be the 'best' simulation? There is a clear problem during the ablation period; is it really a "best" simulation if ablation is too rapid? If stats are calculated throughout the season, and ablation season is short, it is easy to discount the errors during this time of year. But doesn't timing of snow disappearance matter? Perhaps a metric of snow disappearance date should be included? One could argue the result in 5f is much worse than 5d, so that assimilation is not improving the simulations, but actually making it worse.

---

## Author Comment (AC1) · 24 Jan 2021

DUE January 25th, 2021 Reviewer Comments and Responses

Reviewer #1:

The reviewer's comments are preceded by: Comment The authors responses are preceded by: Response
* * *
Comment:

The article is interesting and innovative. The use of data measured by the community is a contribution to the simulation of snow distribution and a way of bringing the community closer to snow science and hydrology. The scientific quality of the article is good; however, the article could improve the analysis on some topics described below.

First, despite mentioning that the distribution of snow by the wind is important, the article does not present results or analysis in this regard. Snowmodel allows you to export the results of wind redistribution. Showing these results would be a contribution to the analysis and discussion.

Response:

SnowTran-3d does allow for variables to be exported for analysis, and these variables include: snow depth (m), saltation transport (m), suspension transport (m), sublimation (m), snow redistribution at the time step (m), summed sublimation (m), and summed blowing snow transport (m). During the calibration of SnowModel for the domain, before measurements were assimilated, we tested the results of SnowModel simulations with SnowTran-3d turned off and with SnowTran-3d turned on. These initial results showed that simulations using SnowTran-3d were consistently outperforming those without it, according to various calibration metrics at the Upper Tsaina SNOTEL location. At this point we determined that we should use SnowTran-3d for all simulations, both for the no assimilation case and the CSO assimilation case.

The wind related variables exported by SnowTran-3d would not be altered by the data assimilation process, since SnowAssim only modifies the precipitation inputs and snowmelt factors, not the wind speed fields or wind direction fields. Additionally, the snow depth variable was exported and analyzed extensively throughout the manuscript, playing a key role in our methodology, validation, and final results. The authors believe that our analysis of the snow depth distributions in the manuscript are sufficient and the decision to use SnowTran-3d as a parameter tested in the calibration was prudent.

[Figure]

———————————————————————————————-

Comment:

Also, a comparison simulation without wind redistribution (Windtrans off) would be a way to measure the improvement of using this tool.

Response:

See the answer above regarding the calibration workflow when we tested results with and without SnowTran-3d. See also Appendix A on line 605 which shows that Snow-Tran3d was tested before assimilation. The authors believe that any further exploration of the SnowTrand-3d sub-model results lies outside the scope of this manuscript, as our research questions are not directly related to the effects of CSO measurement assimilation on wind transportation processes.

———————————————————————————————-

Comment:

Secondly, the assimilation in Snowmodel is highly dependent on swe point location, in addition to timing. It is important to consider in the analysis where the data used are located. And if they agree in time and place with the validation dataset. If the SWE data used for assimilation are located close to the validation point. Logically the result will be very similar to the validation point measure since the model corrects the precipitation or fusion to obtain a value close to the given one. For this reason, it is important to know how close the CSO data is to the field work data. If these two data are very close in time and location it does not make sense to use the field work data for validation.

Response:

The authors agree that it is necessary to be aware of the location of the CSO measurements in space and time in comparison to the validation datasets location in space and time. We provide the following explanation to clarify the location and timing of the CSO

measurements assimilated.

First, the time-series analysis validation metrics were quantified for all days of the water year in both years at the Upper Tsaina SNOTEL location. The CSO measurements that were assimilated in 2017 range in distance from 4.1 km to 30.5 km away from the SNOTEL station location. The CSO measurements that were assimilated in 2018 range in distance from 2.1 km to 17.4 km away from the SNOTEL station location. These distances mean the CSO measurements used in the assimilation do not coincide with the SNOTEL grid cell location and should be included in the section 6.1 time-series validation.

Secondly, the 2018 fieldwork measurements (co-located snow depth and SWE) were used to validate the model results with assimilation. As noted in lines 478-479 in section 6.3, "we separated those [2018 fieldwork] measurement sites used in the assimilation scheme from the validation set when creating Table 3." Due to this, we think the fieldwork measurements and analysis should be included in the section 6.3 fieldwork results.

Thirdly, the remote sensing datasets were collected on April 29th in 2017 and April 7th/8th in 2018. These validation datasets are essentially a spatial snapshot of snow depth from a single day in both water years. In water year 2017, there were a total of 9 CSO measurements submitted on April 29th, the same day as the remote sensing dataset collection. For the presented results in Section 6.2 from the highest performing (Best) simulation with assimilation and the median performing (Median) simulation with assimilation, none of these 9 CSO measurements from April 29th were used. For water year 2018, the remote sensing dataset was collected on April 8th and the measurements were not assimilated in time until at least April 15th (see the experimental design outlined in Section 5 lines 354 to 369 which states that we selected the CSO measurements for assimilation that were collected on or after April 15th of each water year). Due to all of these factors, the remote sensing dataset validation should be included in section 6.2.
Additionally, the remote sensing datasets are distinct, in both form and collection method, from the CSO measurements. All of the analysis in section 6.2 is aggregated to the entire spatial domain of the RS datasets, not at a single point like a CSO measurement location. This fact is why these datasets are important to include in the validation, because they can show the effects of assimilation throughout a complex and variable mountainous terrain.

—————————————————————————————————-

Comment:

Finally, the article should include a comparison between the data used: RS, CSO and field work data. The objective is to check if the data are consistent with each other and if they are very similar in time and location.

Response:

In our submitted manuscript, we did not find it necessary to include analysis comparing the remote sensing datasets, the CSO measurements, and the fieldwork measurements. This is primarily because there is not a single day of measurements that would work to make this comparison between all datasets.

—————————————————————————————————-

Comment:

Also, the article should include a comparison between the densities estimated to convert the CSO data to snow water equivalent and the densities measured in the field work.

Response:

The authors agree with the reviewer that comparing the SWE values measured at the 2018 fieldwork sites to the SWE values estimated by Hill et al. (2019) would add clarity to the results and quantify the uncertainty that is added when converting the CSO snow

depth measurements to SWE. See the following sentences that will be added to section 6.3 Fieldwork Results:

"Additionally, we can use the co-located snow depth and SWE measurements at the fieldwork sites to quantify the uncertainty that is added to the model during the snow depth to SWE conversion. By converting the fieldwork snow depth values to SWE using the Hill et al. (2019) method, we can compare the measured SWE to the approximated SWE values. The fieldwork measurement RMSE in SWE is 10.5 cm and the Bias in SWE is 0.6 cm when using the Hill method for all fieldwork sites."

———————————————————————————————-

Comment:

Some specific comments:

1) Figure 1 and 3 should be next to each other or join them to be able to compare the distribution of the data used for assimilation and validation

Response:

The authors are amenable to combining figures 1 and 3 if the editor or the production design team thinks it's a better use of space or would be easier for the readers to understand. We note that they include different types of data that are introduced in different sections of the manuscript, so keeping them separate may be easier for readers.

———————————————————————————————-

Comment:

2) Point 3.2.5 Snow depth to snow water equivalent conversion. Add the uncertainty in the snow density estimation.

Response:
We changed the following sentence in section 3.2.5 of the submitted manuscript to include the bias and RMSE from Hill et al. 2019.

"Second, it was found to outperform other bulk density methods such as Sturm et al. (2010) and Jonas et al. (2009) when tested against a wide variety of snow pillow and snow course datasets, with an overall bias of 2 mm and RMSE in SWE of 6 cm (Hill et al., 2019)."

————————————————————————————————————-

Comment:

3) Point 6 why the Sugarloaf Mountain station is not used to validate the results?

Response:

We used temperature data from the Sugarloaf SNOTEL station to calculate local lapse rates for the calibration analysis. Since the station does not have snow water equivalence measurements (stated in line 256), we did not use the data for any other purpose. This point could be made more clearly, and we suggest adding the following sentence to the manuscript section 3.4.1.:

"The SLS station data was used to create local temperature lapse rates for the calibration and the UTS station data is used in the manuscript results section to create the SWE time series analysis."

————————————————————————————————————-

Comment:

4) Point 6.2 The location or spatial distribution of CSO measurement used for the assimilation is as important as the number and should be and it should be analyzed here or elsewhere.

Response:
We specifically did not include the spatial distribution of CSO measurements in the research questions of this manuscript. In order to address questions about the spatial representativeness of CSO measurements, we think more extensive fieldwork measurement campaigns or coordinated CSO campaigns would be required. We think that taking regular measurements within a study area across 1) multiple elevational gradients, 2) a broad array of land cover types, 3) a representative sample of slope angles, and/or 4) a representative sample of aspects would help untangle these multiple landscape controls on the spatial distribution of the snowpack. The research design of the current study was not set up to incorporate this type of analysis, however we absolutely agree with the reviewer that this is an interesting and important question moving forward. We conducted some initial spatial analysis of the CSO measurement locations and metric ranking results, and this initial analysis was messy and complex. We note that the CSO modeling team has set up experiments in other locations in the continental U.S to address these various spatial distribution of CSO measurements questions. These include study areas where more measurements have been taken per water year and more SNOTEL stations exist for validation purposes.

---

## Author Comment (AC2) · 24 Jan 2021

DUE January 25th, 2021 Reviewer Comments and Responses

Reviewer #2:

The reviewer's comments are preceded by: Comment The authors responses are preceded by: Response
* * *
Comment

The manuscript describes use of a new citizen science dataset (snow depth) to guide simulations of SWE via data assimilation (DA). The motivation is to include observations gathered from locations in the landscape that might not be monitored otherwise. The study is focused on a maritime snow climate of Alaska. The model used (Snow-Model) has a long history and is well established. A range of different observations (Snotel, field surveys with depth and SWE, and remotely-sensed snow depth) are used to gauge performance of the DA system, compared to simulations that do not include the depth observations.

The results presented are interesting and there appears to be considerable potential for use of the citizen science depth observations. However, major revisions could help make the manuscript more useful. The following issues should be addressed:

First, a more complete description of the DA approach is needed. In the introduction, a more detailed comparison of the approach used relative to other snow DA efforts should be provided – beyond what is currently included in the introduction (e.g., L80). How does the approach used compare to other methods, including direct insertion (e.g. Hedrick et al., 2018), particle-batch smoother (Margulis et al. 2019), particle filter (Smyth et al. 2019) and possibly EnKF.

Response:

The authors do not think this article needs to review all of the data assimilation methods used in snow science in great detail (particle filters, particle batch smoothers, Kalman filters, and ensemble Kalman filters). However, we understand the reviewer's desire to add context to the manuscript regarding other types of data assimilation methods. We propose the following paragraphs to replace the single paragraph in the methods section 3.2.4, since these new paragraphs more clearly describe the way SnowAssim works and they compare SnowAssim to other assimilation methods.

[revised manuscript text omitted]

pp.1752-1772.

————————————————————————————–

Comment:

The methods section provides only a limited description of how the model is adjusted for mismatch with observations ("SnowAssim aggregates all the assimilated observations by date and creates a spatially varying cor- rection surface that covers the entire model domain (Liston and Elder, 2008). These various correction surfaces are applied by adjusting the model precipitation fluxes and snowmelt factors between SWE observation dates during a second SnowModel simulation"). The 'adjustments' to the model are central to the effort, so the method should be described more completely in the manuscript.

Response:

The authors note that the literature on SnowAssim (Liston and Heimstra, 2008) is cited in the methods section 3.2.4 and we make efforts to not repeat information from that publication. First, see Figure RC1 (Figure 6a from Liston and Heimstra, 2008) included in our response below for the reviewer's ease, as an example of a correction surface referenced above. We do not include an example of the correction surface in the manuscript because it is explained in the original literature. However, see the previous answer for additional information that will be added to the manuscript's method section 3.2.4.

————————————————————————————–

Comment:

The results (or discussion) do not include any documentation of the 'adjustments' to the model, yet one of the benefits of DA is that the merging of data and models is one way to more completely understand the entire system (e.g., see Magnusson et al., 2014 and 2017 and their retrieved precipitation correction factor).

Response:

The authors agree that including some additional information from the correction factor adjustments during assimilation would benefit the arguments we make in the manuscript and elucidate the entire modeling/assimilation system. See Table RC1, which includes data from the best ranked assimilation runs using the time-series and spatial analysis. We plan to add the following paragraph to a new section (6.6) in the results.

" 6.6 Correction Factor Results SnowAssim generates a set of correction factors for each of the CSO ensemble member simulations. These factors correspond to the observed and measured differences in the SWE variable and are used to create a correction surface with the Barnes objective analysis. Table [RC1] reviews a subset of the correction factors, including data from the Best ranked CSO simulations according to the various temporal and spatial metrics previously reviewed in sections 6.1 and 6.2. The number of observations varies for the Best ranked simulation, as well as the precipitation correction factors, the use of a melt correction factor, and whether or not an interpolated correction surface was created. These correction factor results show that relatively few measurements are needed during assimilation and that there are multiple paths to improving model performance when assimilating CSO observations using SnowAssim."
* * *
Comment:

Second, uncertainty of the observations and validation data should be described and incorporated into the analysis. One of the benefits of DA is that the magnitude of uncertainty can be explicitly included in the analysis (e.g., Magnusson et al., 2014 and 2017). It appears that uncertainty of the assimilated observations is not included in the analysis – is this the case? If not, why not?

[Figure]

The spatial representativeness of the depth measurements is mentioned in the discussion. One component of uncertainty is related to the conversion from depth to SWE, using the density estimation described in Hill (2019). In the region analyzed, SWE estimates based on density from Hill (2019) have an RMSE of 0.2-0.25 (normalized to snow season precipitation). Is this considered in the DA approach? Uncertainty (or biases) of the validation data is not described, thus it is implied that the data are 'perfect'. What is the error or uncertainty associated with the federal sampler data?

Response:

The authors think that the reviewer brings up an important point about the need for further uncertainty analysis in the manuscript. We note that SnowModel is a deterministic model and SnowAssim does not include an explicit characterization of uncertainties, so any perturbations need to be based on fieldwork measurements or other previously reported error values. We took this opportunity to characterize multiple sources of uncertainty mentioned by the reviewer in Table RC2.

One of the sources of uncertainty mentioned by the reviewer comes from the conversion of snow depth to SWE using the Hill et al. (2019) method. We can quantify this source of error using the reported values from Hill et al. (2019) or using the fieldwork measurements of co-located snow depth and SWE (see Table RC2 above). Another source of uncertainty is the spatial representativeness of the depth measurements across short distances. The average standard deviation of snow depth at all 70 fieldwork sites is 22 cm of snow depth, and after conversion to SWE, the average is 8.7 cm of SWE. We can assume that the spatial variability of snow depth plays a role in the conversion method uncertainty, so we decided to choose one of these values and perturb SWE measurements used in the data assimilation scheme. In an attempt to be conservative with our error estimates, we chose the highest reported/measured error value of 10.5 cm to create an envelope of uncertainty around our SWE values reported in the assimilation runs at the Upper Tsaina SNOTEL station. Figure RC2 contains the results of these additional model runs. This uncertainty analysis displays improved

simulation results, even after the error estimation from these sources of uncertainty have been taken into consideration. This figure further contextualizes the temporal results in section 6.1 and we suggest adding it to the manuscript with the following sentences.

"Using the snow depth to SWE conversion method during assimilation introduces uncertainty into the modeling process. Instead of using the global estimates of error reported in Hill et al. (2019; RMSE in SWE = 5.9 cm) we decided to calculate this source of error using our fieldwork site measurements. The RMSE in SWE due to the conversion method is 10.5 cm and we perturbed the CSO observations by this amount to depict the upper and lower boundaries of error associated with this source of uncertainty. Figure [RC2] displays the Best CSO simulation temporal results for each WY, along with the UTS station SWE record and the NoAssim case. These perturbations to the assimilated SWE show improved modeled SWE values at the UTS station when compared to the NoAssim case, even after this source of uncertainty has been accounted for."

Since there is additional uncertainty associated with the federal sampler data, we decided to add the following sentence to the methods section 3.2.3:

"Federal sampler data collection introduces uncertainty in the form of measurement error due to variable snow conditions and densities, hard impenetrable crusts, and loss during extraction. Dixon and Boon (2012) report the results of several studies showing that the Federal Sampler error, as a percentage of SWE depth, ranges from 4.6% to 11.2%. Our results presented in Section 6.4 include field measurements of SWE that use the higher 11.2% value for conservative SWE error estimation."

Because of this uncertainty we modified Table 4 in line 510 (Section 6.4) of the submitted manuscript to include a range of estimated densities and SWE (+/- 11.2%) instead of using only the measured values without the error estimation. See Table RC3 for the new values that will be added to the original Table 4 in red.

Lastly, we know there is uncertainty associated with the remote sensing acquisitions, and the sources of this error include flight trajectory and geometry, laser scan angle, density of canopy, steep gradients in the terrain, and more (Deems and Painter, 2006). We decided to report the estimated uncertainty for the RS datasets in the methods section 3.4.2 and also include them in the new version of Table 4. The following sentences will be added to the methods section 3.4.2:

"There is uncertainty associated with the RS dataset acquisitions, and the sources of error are related to flight trajectory and geometry, laser scan angle, density of vegetation and canopy, and steep gradients in the terrain (Deems and Painter, 2006). The mean error in snow depth for the photogrammetry and LiDAR datasets are estimated at 10.4 cm and 1.1 cm, respectively. The vertical RMSE in snow depth are estimated at 31.0 cm for the photogrammetry and 10.2 cm for the LiDAR dataset. While we acknowledge and report these error estimations, they are integrated into the results in Table 4 in Section 6.4 but not used in the spatial results reported in Section 6.2."

Dixon, D. and Boon, S., 2012. Comparison of the SnowHydro snow sampler with existing snow tube designs. Hydrological Processes, 26(17), pp.2555-2562.

Deems, J.S. and Painter, T.H., 2006, October. Lidar measurement of snow depth: accuracy and error sources. In Proceedings of the 2006 International Snow Science Workshop: Telluride, Colorado, USA, International Snow Science Workshop (pp. 330-338).
* * *
Comment:

Third, something seems strange about the calibration and validation methods and results. Are the NSE values in Table 1 correct? If the best simulation has NSE < 0, this would suggest that the calibration is not working very well. Additional details are required.

Response:

The values in Table 1 in the original manuscript are correct and they were the impetus for including the precipitation adjustment experiment in the manuscript. The initial calibration model runs and the final No Assimilation model runs were displaying timeseries SWE values that were consistently high, throughout both water years, with both reanalysis products (see original Figure 5a and 5d). We know that biases in meteorological forcings are one of the most important factors in estimating snow depth and SWE magnitudes correctly (Liston and Heimstra, 2008; Margulis et al., 2015). So we decided to take a closer look at the precipitation totals with the CFSv2 product. See Figure RC3 that shows the total amount of precipitation over the calibration period when compared to the Upper Tsaina Snotel station and compared to when a precipitation adjustment factor is used. The authors would like to add this figure to the appendix to clarify the need for precipitation adjustments, whether via data assimilation or the precipitation adjustment experiment.

Because of this bias in the meteorological inputs, and after a conversation with the model developer about the calibration challenges in this region of Alaska, the authors were confident that making adjustments to the model parameters would only slightly improve our snow depth and SWE distributions and magnitudes (see Appendix A for a full list of the model parameter adjustments made during calibration). The improvements that could be made by adjusting model parameters were insignificant when compared to adjusting the precipitation fields.

Importantly, the reviewer's question speaks directly to why we think SnowAssim is the correct assimilation method for this research. SnowAssim adjusts the precipitation fluxes and/or snowmelt factors using only the additional observations provided by CSO participants. Nothing else is changed and no additional information is required for this type of data assimilation. Recall that we are not forcing the model with in-situ weather station data because the required meteorological variables are not available within the domain. Even with biased and coarse reanalysis forcing data, SnowModel

Interactive
comment

and SnowAssim are able to make snow depth and SWE magnitude improvements by the simple addition of several in-situ snow depth observations, strengthening our key claim that "that even modest measurement efforts by citizen scientists have the potential to improve efforts to model snowpack processes in high mountain environments."

Liston, G.E. and Hiemstra, C.A., 2008. A simple data assimilation system for complex snow distributions (SnowAssim). Journal of Hydrometeorology, 9(5), pp.989-1004.

Margulis, S.A., Girotto, M., Cortés, G. and Durand, M., 2015. A particle batch smoother approach to snow water equivalent estimation. Journal of Hydrometeorology, 16(4), pp.1752-1772.

———————————————————————————————-

Comment:

Is calibration for the entire year? The entire snow year? Why not at peak SWE?

Response:

As mentioned in lines 308 to 309 in the submitted manuscript, the calibration time period is for the entire water year, for 5 years. The calibration statistics cover the entire 5 year period. Two months (July and August) of data per water year, in which no snow was modeled, measured, or expected at the Upper Tsaina SNOTEL station in the domain, were removed from the calibration metrics. This was in an attempt to not bias the results of the RMSE and mean bias error metrics with months of corresponding zeros from the observed and modeled vectors.

———————————————————————————————-

Comment:

Results in Fig 5 also seem strange. Fig 5e: how can this be the 'best' simulation? There is a clear problem during the ablation period; is it really a "best" simulation if ablation is too rapid? If stats are calculated throughout the season, and ablation season is

short, it is easy to discount the errors during this time of year. But doesn't timing of snow disappearance matter? Perhaps a metric of snow disappearance date should be included? One could argue the result in 5f is much worse than 5d, so that assimilation is not improving the simulations, but actually making it worse.

Response:

Figure 5e represents the data from the Best ranked simulation according to the time-series data from water year 2018. The decision-making for characterizing the Best ranking is explained in lines 381 through 383 and also qualified in the discussion section, in lines 536 through 542 of the submitted manuscript. Characterizing and focusing on the Best results for some figures in the manuscript was a decision made by the authors to show a sampling of the many model runs we conducted during the analysis, instead of overwhelming the reader with too many figures. We acknowledge that if we focused on just the accumulation phase or just the ablation phase, the characterization of the Best results would indeed look different. The ablation phase of the snowpack in Figure 5e is not a perfect match to the 1:1 identity line, but the corresponding metrics (NSE, KGE, RMSE, Bias) all show improvements when compared to the No Assimilation case when averaging over the water year. The new SWE figure that includes error perturbations shows that the timing of the last day of SWE (snow disappearance date) in the Best CSO simulation in WY2018 is 6 days earlier than the SNOTEL snow disappearance date. The range of snow disappearance dates when accounting for some level of measurement and conversion model uncertainty is from 10 to 1 day(s) early. The NoAssim snow disappearance date is 7 days later than SNOTEL. The WY2017 snow disappearance dates are even better.

The claims that we make in the results and discussion section are specific to the entire water year and we are careful to not make any claims about improving the snow disappearance dates or the timing of the melt period. However, after doing additional uncertainty analysis as suggested by the reviewer, we are more confident that our overall snow disappearance dates are acceptable. The authors also note that the magnitude of peak SWE is greatly improved in our best model runs when compared to the NoAssim case, and this may be more important for readers concerned with the water resources implications of our work. While we acknowledge that there are alternate ways to subset the data temporally, the authors stand by our decision to use water year averaged metrics to characterize the Best ranked simulation.

Lastly, since the reviewer requested, we plan to add the following sentences to the new paragraph accompanying the new SWE figure in section 6.1 in order to report the Best CSO simulation snow disappearance dates:

"Since the timing of snow disappearance is important for ecological systems in alpine environments and water resources managers, we calculated the range in snow disappearance dates from the best simulations from both water years. In WY2017 and WY2018, the snow disappearance date for the NoAssim case is 10 and 7 days later than the UTS station, respectively. In WY2017, the snow disappearance date in the Best CSO simulation, accounting for measurement uncertainty, ranges from 3 days early to 8 days later than the UTS station. In WY2018, the range is from 10 days to 1 day earlier than the UTS station. These ranges in snow disappearance date are acceptable and show improvements in model performance for some, but not all, of the Best CSO simulations after accounting for measurement uncertainty."

―――――――――――――――――

[Figure]

*Figure RC1: This is Figure 6a from Liston and Heimstra (2008) showing an example of a spatially distributed correction factor surface created by the SnowAssim data assimilation scheme. These values modify the 1) precipitation inputs during the accumulation phase or 2) the melt rates during the ablation phase. We do not intend to add this figure to the manuscript.*

**Fig. 1.** Figure RC1

[Figure]

Figure RC2: Snow water equivalent (SWE) time series results with measurement uncertainty included. The simulations with ±10.5 cm of SWE represent the upper and lower boundaries of error introduced when converting snow depth measurements to SWE using the Hill et al. (2019) method.

**Fig. 2.** Figure RC2

[Figure]

*Figure RC3: Precipitation totals at the Upper Tsaina SNOTEL station compared to the CFSv2-forced model totals and the CFSv2-forced model totals with a precipitation adjustment factor. This overestimation of precipitation by the reanalysis product is a major factor in the quality of the calibration results.*

**Fig. 3.** Figure RC3

*Table RC1: Correction factors from the assimilation scheme for the best ranked simulations from both water years. The model determination for precipitation vs melt correction factors is included and whether or not the Barnes objective analysis created a spatially distributed correction surface.*

| Type | Ranking | Year | # of Obs | Precip Correction Factors | Melt Correction Factors (-) | Interpolated Surface? | Dates |
|---|---|---|---|---|---|---|---|
| Temporal | Best | 2017 | 2 | 0.45, 1.04 | n/a | Yes | 4/29/17 |
| Temporals | Best | 2018 | 2 | 0.68, 0.76 | n/a | Yes | 5/15/18 |
| Spatial | Best | 2017 | 8 | 0.30, 0.50, 0.73, 0.86, 1.36 | 6.32, 2.29, 22.6 | Yes | 4/29/17; 5/8/17 |
| Spatial | Best | 2018 | 1 | 0.32 | n/a | No | 5/22/18 |

**Fig. 4.** Table RC1

*Table RC2: Sources of uncertainty, calculated or reported error, and metric used for each dataset. We do not intend to add this table to the manuscript, it is included for the reviewer only. All of the data will be included in various locations within the final, edited manuscript.*

| Source of Uncertainty | Error | Metric |
|---|---|---|
| Conversion Method (reported) | 5.9 cm | RMSE |
| Conversion Method (measurements) | 10.5 cm | RMSE |
| Variability of Snow Depths (measurements) | 8.7 cm | Std. Dev. Ave. |
| Federal Sampler Measurement Error (reported) | 11.2% | % Error |
| 2017 Photogrammetry RS Dataset (measurements) | 10.4 cm, 31.0 cm | Mean Error, RMSE |
| 2018 LiDAR RS Dataset (measurements) | 1.1 cm, 10.2 cm | Mean Error, RMSE |

**Fig. 5.** Table RC2

*Table RC3: Spatially Averaged Variables in the RS Region*

*The spatially averaged results were calculated using the RS region in WY2018, the RS dataset (±1cm error), the spatially averaged density, and the modeled results. The spatially averaged SWE depth for the RS survey was estimated using the average density (± 11.2%) measured during April 2018 fieldwork.*

| Dataset | Spatially Averaged Snow Depth (cm) | Spatially Averaged Density (kg/m³) | Spatially Averaged SWE Depth (cm) | Total RS Region Water Volume (km³) |
|---|---|---|---|---|
| RS Survey 2018 | 130 ±1 (RS survey) | 331 ± 37 (fieldwork) | 38 - 48 (estimated) | 0.06 - 0.07 (estimated) |
| Best CSO Simulation 2018 | 130 (modeled) | 400 (modeled) | 52 (modeled) | 0.08 (modeled) |
| NoAssim 2018 | 267 (modeled) | 430 (modeled) | 115 (modeled) | 0.17 (modeled) |

**Fig. 6.** Table RC3

---

## Referee Report (RR1)

**Hydrology and Earth System Sciences**
Review of Manuscript hess-2020-321

**Summary**

The authors present a method for assimilating snow depth observations collected by citizen scientists into a process-based snow model, which improves snow depth, mass, and disappearance date estimates relative to model runs without assimilation. The model outputs are improved regardless of the number of assimilated observations. The paper is well-written. I recommend the paper for publication after addressing the following comments and suggestions.

**Major Comments**

1) Why is the snow depth to SWE conversion necessary before assimilation? It is unclear if assimilating SWE is a literal requirement of the SnowAssim submodel, or if you are making a choice to assimilate SWE rather than assimilating snow depth directly. Previous studies (e.g., Smyth et al., 2019) have shown that assimilating snow depth improves modeled depth, and can also improve modeled density and SWE. I am not asserting that assimilating depth is better than assimilating SWE, but I think the paper needs a justification for including the depth to SWE conversion (and its associated error, even if minimal) as part of the workflow at all.

2) I was going to comment on the negative NSE values in table 1 (model calibration results), and then saw that a previous reviewer already did so. I think the authors' response makes sense, and the supplemental experiment with different precipitation adjustment factors is logical. However, I have two suggestions:
- The sentence on lines 361-362 should be more clear. Without reading your response to the previous reviewer, I would not understand that your model doesn't have a "snowfall correction factor" parameter that could be calibrated, as some other models do. Otherwise, a reader would probably not be satisfied that you simply say that NSE is "lower than expected." A negative NSE implies that you would be better off throwing away your model and using the mean of the observed data, which is a poor starting point for a supposedly calibrated model. Again, I realize that the model IS calibrated at this point, and the precipitation data is just terrible, but that point doesn't come across clearly in the paper.
- While the experiment with the precipitation adjustment factors is sufficient from my point of view, it sure would be a lot cleaner if you simply added a "precipitation correction factor" to your model and calibrated it along with everything else. Is that feasible? As you say, one of the benefits of your assimilation framework is that it "fixes" this bad-precipitation problem for you. But that also means you are choosing to calibrate some things, and choosing to let the assimilation fix the rest – which again, does not come through clearly in the text until later in the results.

3) I am confused by the methodology and conclusions relating to the experiments where the authors vary the number of CSO observations that are assimilated. I understand that the number of assimilated observations is varied between 1 and 32. As fewer observations are assimilated, the model receives less information – but are we talking about restricting the amount of information across space, time, or both? When you say that "Any number of CSO measurements

assimilated show improvements in model performance" (429) do you mean that any number AND any timing of assimilated observations leads to improvements? Or again, on line 462, where you say that "WY2017 has a smaller range in KS values as the number of assimilated measurements increases, more CSO simulations outperforming the NoAssim case" – is this because "more CSO simulations" cover a larger geographic area, or because they cover a longer time period? (or both)

- I have a related (more minor) question on line 391: what does "aggregated by week" mean? You assume all observations in a given week occurred at the same time, for the purposes of assimilation?

**Minor Comments**

82-84: A related goal (at least for statistical assimilation methods like the Kalman Filter, Particle Filter etc.) is to reduce the *uncertainty* of the given state variable.

109: This is the first mention of SnowAssim, and it is not defined/explained yet. Maybe simply omit the reference to the specific name here?

121-123: I was wondering when the motivation for using CSO depths (as opposed to lidar, etc.) would be mentioned in the introduction. It feels like this sentence belongs somewhere in the paragraph starting on line 86.

240: "State" missing an "s"

Figures 1 and 3: I agree with a previous reviewer that these figures should be combined (side-by-side).

Appendix C: Consider adding SNOTEL SWE to the plot, as another data point. For example, hopefully the measured cumulative precipitation is not less than measured peak SWE, indicating undercatch, etc.

---

## Author Response (AR2)

**Responses to Reviewers**

The following responses to Reviewer #3 and Reviewer #4 include their comments and a point by point response by the authors, which is highlighted by bold font.

**Reviewer #3**

Summary
The authors present a method for assimilating snow depth observations collected by citizen scientists into a process-based snow model, which improves snow depth, mass, and disappearance date estimates relative to model runs without assimilation. The model outputs are improved regardless of the number of assimilated observations. The paper is well-written. I recommend the paper for publication after addressing the following comments and suggestions.

Major Comments

1) Why is the snow depth to SWE conversion necessary before assimilation? It is unclear if assimilating SWE is a literal requirement of the SnowAssim submodel, or if you are making a choice to assimilate SWE rather than assimilating snow depth directly. Previous studies (e.g., Smyth et al., 2019) have shown that assimilating snow depth improves modeled depth, and can also improve modeled density and SWE. I am not asserting that assimilating depth is better than assimilating SWE, but I think the paper needs a justification for including the depth to SWE conversion (and its associated error, even if minimal) as part of the workflow at all.

Include a couple sentences about the density variation found in the fieldwork.

**Response: The snow depth to SWE conversion is required because a SWE depth (m) is the input for the physical equations in the model code that govern the assimilation process SnowAssim (Liston and Heimstra, 2008, their section 3, paragraph 1). Assimilating snow depth would have required altering the model code and physical equations, and this was outside of the scope of our research questions. In Smyth et al. (2019) a different snow model, Snobal, is used with a particle filter to assimilate gridded snow depth data. This is a fundamentally different approach to data assimilation and snow dataset acquisition.**

**The authors have decided to alter the following sentence in line 214-215 of the current manuscript to further clarify that the model requires SWE depth for assimilation:**

**"Note that CSO measurements are submitted as snow depth (m), but the SnowAssim model code and physical equations require observational inputs to be SWE depth (m), so a conversion from depth to SWE was necessary."**

2) I was going to comment on the negative NSE values in table 1 (model calibration results), and then saw that a previous reviewer already did so. I think the authors' response makes sense, and the supplemental experiment with different precipitation adjustment factors is logical. However,

I have two suggestions:

- The sentence on lines 361-362 should be more clear. Without reading your response to the previous reviewer, I would not understand that your model doesn't have a "snowfall correction factor" parameter that could be calibrated, as some other models do. Otherwise, a reader would probably not be satisfied that you simply say that NSE is "lower than expected." A negative NSE implies that you would be better off throwing away your model and using the mean of the observed data, which is a poor starting point for a supposedly calibrated model. Again, I realize that the model IS calibrated at this point, and the precipitation data is just terrible, but that point doesn't come across clearly enough in the paper.

- While the experiment with the precipitation adjustment factors is sufficient from my point of view, it sure would be a lot cleaner if you simply added a "precipitation correction factor" to your model and calibrated it along with everything else. Is that feasible? As you say, one of the benefits of your assimilation framework is that it "fixes" this bad-precipitation problem for you. But that also means you are choosing to calibrate some things, and choosing to let the assimilation fix the rest – which again, does not come through clearly in the text until later in the results.

**Response: The authors have attempted to address multiple important points brought up by the reviewer. In the calibration section (Section 4), we added several sentences that aim to clarify our decision making process during calibration and highlight the deficiencies in the precipitation inputs. We also added text that paraphrases our response to a previous reviewer that requested more information about the calibration results. This was added so the readers have easy and quick access to the substance of this conversation without needing to look into the comment discussion section. We tried to more clearly communicate that we chose to let the assimilation of CSO measurements address the precipitation deficiencies. Please see the text below that has been added to the calibration section of the current manuscript (Section 4, lines 372-386):**

**"CFSv2 precipitation totals at the UTS station were nearly 1.6 times the measured precipitation at the UTS station during the calibration period. The improvements that could be gained by adjusting a subset of the model parameters (wind, temperature, and precipitation lapse rates due to differences in elevation) during calibration were not likely to overcome this extreme precipitation deficiency, explaining why the final calibrated NSE and KGE values were low. There are two ways to address this precipitation deficiency using SnowAssim. One is to adjust the precipitation inputs during calibration, and the other is to allow the assimilation to adjust the precipitation inputs. Both ways are functionally equivalent because they apply a simple, scalar-based correction surface to the precipitation fluxes. In our calibration process we chose to use SnowAssim to address the precipitation deficiencies in the reanalysis product, following the approach of other recent studies in mountainous regions of Alaska, and following the original purpose of the SnowAssim model (Cosgrove et al., 2021, their Calibration of SnowModel section; Liston and Heimstra, 2008; Young et al., 2020, their section 3.4). This calibration decision supports the primary goal of the current study, which is to test whether or not participant-submitted snow depth measurements can improve physically-based modeling efforts through data assimilation.**

**These calibration results and the precipitation deficiencies motivated us to design an experiment to supplement the main findings of this research … ”**

**In our response and additional text above, we have attempted to more explicitly communicate why we chose to let the assimilation address the precipitation deficiencies. Since the goal of the current study is to test whether or not participant-submitted snow depth measurements can improve physically-based modeling through data assimilation, we think our choice to show the results (Sections 6.1 to 6.5) without a precipitation correction is reasonable. We ran additional calibration statistics for the precipitation correction factor experiment results (Section 6.6) and we have a full ensemble of runs and results that correspond to this experiment. However, we chose not to show both sets of results side-by-side throughout the results section for the sake of simplicity.**

**We think the choice to keep the calibration process simpler is defensible because it requires less *in-situ* weather station data. This is a real-world data challenge that affects water resources managers and scientists working in remote, mountainous terrain worldwide. We think the results that include a snow model, a simple data assimilation scheme, coarse reanalysis product forcing datasets, and modest citizen scientists' efforts is more accessible and more likely to be applied by readers than a study that requires high quality, *in-situ* weather station data.**

**Cosgrove, C.L., Wells, J., Nolin, A.W., Putera, J. and Prugh, L.R., 2021. Seasonal influence of snow conditions on Dall's sheep productivity in Wrangell-St Elias National Park and Preserve. PloS one, 16(2), p.e0244787.**

**Young, J.C., Pettit, E., Arendt, A., Hood, E., Liston, G.E. and Beamer, J., 2020. A changing hydrological regime: Trends in magnitude and timing of glacier ice melt and glacier runoff in a high latitude coastal watershed. Water Resources Research, p.e2020WR027404.**

3) I am confused by the methodology and conclusions relating to the experiments where the authors vary the number of CSO observations that are assimilated. I understand that the number of assimilated observations is varied between 1 and 32. As fewer observations are assimilated, the model receives less information – but are we talking about restricting the amount of information across space, time, or both? When you say that "Any number of CSO measurements assimilated show improvements in model performance" (429) do you mean that any number AND any timing of assimilated observations leads to improvements?

**Response: We agree that there is some complexity in our methods and that this section would benefit from some additional clarification. We have added text to explain that the temporal subsetting of CSO measurements is the same throughout the results section, and includes only observations that occur after April 15th. We also explain that we do not restrict the spatial locations of the assimilated measurements, rather they just need to be located somewhere within the model domain. The temporal restrictions are simply that all results shown in the results section used CSO measurements that were taken on or after April 15th of both water years. We added**

several sentences to Section 5: Experimental Design and slightly reordered the presentation of these sentences. See new Section 5 below (lines 395-415 of the current manuscript):

"We carried out a series of simulations in order to (1) quantify the improvement in model performance due to the assimilation of CSO measurements and to (2) understand the effects of the number of CSO data points selected for assimilation. First, we set up geographic and temporal requirements for the assimilated data. The only geographic requirement was that the CSO measurements must be located within the larger 5,736 km2 model domain. We subset the CSO measurements temporally to the peak SWE time period or later. According to the UTS station, peak SWE in the study area generally occurs mid- to late-April and consequently the earliest assimilation date was set to April 15th. The CSO measurements were aggregated by week by assuming all measurements in a given week occurred on the same day for the purposes of assimilation. This weekly aggregation allows the correction surfaces generated by SnowAssim time to adjust the precipitation fluxes and snowmelt factors between observations, thereby altering the model outputs during assimilation. Additionally, CSO participation in the Thompson Pass region during the early accumulation season was infrequent in WY2018 and non-existent in WY2017. Since peak SWE is important for mountain hydrology and ecology, with many snow studies using it as an indicator metric, the time restrictions are acceptable for the research questions addressed in this study (Bohr and Aguado, 2001; Trujillo et al., 2012; Kapnick and Hall, 2012; Mote et al., 2018; Wrzesien et al., 2017).

With these geographic and temporal filters defined for assimilation, we decided to vary the number of CSO data points selected for assimilation. Model simulations without CSO measurements provide a baseline for comparison, referred to as the NoAssim case. Ensemble model simulations were carried out with various numbers of CSO measurements assimilated, referred to as the CSO simulation case. An ensemble of 60 trials per year were carried out with n = 1, n = 2, n = 4, n = 8, n = 16, and n = 32, where n equals the number of CSO measurements assimilated per WY. In each instance (n value), 10 realizations of the numerical experiment were carried out. With the ensemble model simulations defined in terms of the spatial and temporal restrictions, the number of CSO measurements was the only feature modified during assimilation."

Lastly, the authors note that there is a new section in the results (Section 6.3: Spatial and Temporal Characteristics of the Assimilated Data, lines 538-566 of the current manuscript) that includes more details about 1) the spatial characteristics of the assimilated measurements that were selected for the time-series validation analysis and 2) the temporal characteristics of the assimilated measurements that were selected for the remote sensing validation analysis.

Or again, on line 462,where you say that "WY2017 has a smaller range in KS values as the number of assimilated measurements increases, more CSO simulations outperforming the NoAssim case" – is this because "more CSO simulations" cover a larger geographic area, or because they cover a longer time period? (or both)

Response: With the clarification in the answer to the previous question and the new Section 5 in the manuscript, the reviewer's question regarding geographic area and longer time period may be

clarified. However, the authors note that the wording of the referenced sentence is unintentionally vague and should be reworded for further clarity. A more precise sentence has been added to the current manuscript (lines 481-483) and can be found below:

"However, WY2017 has a smaller range in KS values as the number of assimilated measurements increases. Additionally, the number of simulations that outperform the NoAssim case in WY2017 gradually increases as the number of CSO measurements increases from 1 to 32."

The exact cause of this increase in performance in WY2017 with more CSO measurements is not known or investigated by the research questions in the current study, although it is admittedly one of the most interesting areas for future research. One may speculate that incorporating 32 measurements, that span multiple days and multiple geographic locations, would be better than incorporating a single measurement on a single day and in a single location. But the results show that assimilating a single measurement sometimes is just as advantageous as assimilating 32 measurements. There are a multitude of terrain controls on the spatial distribution of snow in mountain environments, including aspect, prevailing wind direction, elevation, curvature (concavity or convexity), etc. These all play a role in geographic location and require a different study design and a more in-depth investigation into the spatial characteristics of the study area than fits within the scope of the current study.

- I have a related (more minor) question on line 391: what does "aggregated by week" mean? You assume all observations in a given week occurred at the same time, for the purposes of Assimilation?

Response: Yes, aggregated by week means we assume all observations in a given week occurred at the same time for the purposes of assimilation. We agree that sentences here were potentially vague and needed to be clarified. We chose this type of weekly aggregation because of the way SnowAssim adjusts precipitation fluxes / snowmelt factors between observation dates. Correction factors for precipitation / snowmelt are applied to the time period between observations (Liston and Heimstra, 2008, their section 3). Our initial model runs suggested that daily increments (consecutive SWE observations) did not allow enough time between observations for the correction factors to influence the SWE outputs. We aggregated all CSO measurements to the same day of the week to allow the correction factors time to adjust precipitation and snowmelt, thus altering the SWE depth outputs during assimilation. We changed the text of the current manuscript (lines 400-403) to add clarification.

"The CSO measurements were aggregated by week by assuming all measurements in a given week occurred on the same day for the purposes of assimilation. This weekly aggregation allows the correction surfaces generated by SnowAssim time to adjust the precipitation fluxes and snowmelt factors between observations, thereby altering the model outputs during assimilation."

Minor Comments
82-84: A related goal (at least for statistical assimilation methods like the Kalman Filter, Particle Filter etc.) is to reduce the uncertainty of the given state variable.

**Response: The authors agree that the reviewer adds an important extension to this sentence and we have changed the current manuscript (lines 83-85) accordingly:**

**"Regardless of the method of assimilation, the goal is the same: to produce a more accurate modeled state variable (snow depth or SWE) in space and time and to reduce uncertainty in the state variable by using in-situ observations to modify the process model output."**

109: This is the first mention of SnowAssim, and it is not defined/explained yet. Maybe simply omit the reference to the specific name here?

**Response: This is a good suggestion and we have changed the text in the current manuscript (lines 111-113) to the following sentence:**

**"The CSO project adds to a growing body of research accomplished by citizen scientists in the natural sciences, and demonstrates how CSO measurements can be assimilated into the process model workflow using a simple data assimilation technique to sometimes improve model results."**

121-123: I was wondering when the motivation for using CSO depths (as opposed to lidar, etc.) would be mentioned in the introduction. It feels like this sentence belongs somewhere in the paragraph starting on line 86.

**Response: The authors agree that the following sentence should be placed earlier in the introduction, and we added it to the current manuscript at lines 88-90.**

**"The potential of mobilizing a new type of *in-situ* snow dataset collected by snow professionals and snow recreationists is significant because these participants often travel to remote mountainous environments worldwide where *in-situ* snow observations are sparse."**

240: "State" missing an "s"

**Response: This change has been made in the current version of the manuscript.**

Figures 1 and 3: I agree with a previous reviewer that these figures should be combined (side-by-side).

**Response: These figures have been combined in the current version of the manuscript.**

Appendix C: Consider adding SNOTEL SWE to the plot, as another data point. For example, hopefully the measured cumulative precipitation is not less than measured peak SWE, indicating undercatch, etc.

**Response: We've added the SNOTEL SWE vector to Appendix C in the current draft of the manuscript.**

**Reviewer # 4**

This is a fairly polished manuscript that has already been through one round of reviews. Following comments from Reviewer #1 on the time and location of assimilation in relation to validation, the authors provide some useful information in their response. This information is therefore now available in the discussion, but it would be better to incorporate it in the paper.

**Response: After reviewing our previous response to Reveiwer #1, we agree with Reviewer #4 that including additional information about the time and location of the assimilation data points in relation to the validation datasets adds important context for the readers. We've adapted our original response to Reviewer #1 into several paragraphs and a new figure in a new Section 6.3. These new paragraphs and figure are below (lines 538-536 in the current manuscript):**

**"The geographic locations of the CSO measurements used in the temporal and spatial results are an important factor that can shed some light on our understanding of the assimilation process. First, the time-series analysis validation metrics were quantified for all days in the water year at the UTS location. The CSO measurements that were assimilated in 2017 range in distance from 4.1 km to 30.5 km away from the UTS location, while the Best CSO simulation measurements (n=2) were located 5.5 and 6.9 km away. In 2018 the assimilated measurements range in distance from 2.1 km to 17.4 km away from the UTS location, and the Best CSO simulation measurements (n=2) were located 9.1 and 17.5 km away. Figure 10 includes a map of the assimilated measurements and a histogram of the distance between the CSO measurements and the UTS station from both water years, subset by the assimilation time period (on or after April 15th of each year). This distance analysis demonstrates that the CSO measurements used in the time-series assimilation do not coincide with the SNOTEL grid cell location. The histogram shows that improvements made at the SNOTEL location during assimilation were due to snow depth measurements taken by CSO participants kilometers away.**

[Figure]

**Figure 10: Assimilated measurements.**
**(a) A histogram showing the distance between the CSO measurements available for assimilation and the Upper Tsaina SNOTEL station, subset by the assimilation time period, on or after April 15th (n=266). A kernel density estimator is used to smooth the distribution. (b) A map of the CSO measurement locations that includes the best spatial and temporal CSO simulations for both water years. The map is zoomed in on the area of the highest density of CSO measurements.**

**Secondly, the remote sensing datasets were collected on April 29th in 2017 and April 7th and 8th in 2018. These validation datasets are essentially a spatial snapshot of snow depth from a single day in both water years. In water year 2017, there were a total of 9 CSO measurements submitted on April 29th, the same day as the remote sensing dataset collection. For the presented results in Section 6.2, none of these 9 CSO measurements from April 29th were used. For water year 2018, the remote sensing dataset was collected on April 8th and the measurements were not assimilated temporally until at least April 15th (see the experimental design outlined in Section 5). Figure 10b displays the locations of the CSO measurements assimilated in the Best CSO simulation from both water years (WY2017 n=1; WY2018 n=8). This analysis of the assimilated data demonstrates that the CSO measurements used in the spatial assimilation do not coincide with the dates of the remote sensing acquisition, revealing that improvements were made during assimilation by measurements that were taken at a different time."**

Related to this, I agree with the reviewer's suggestion of putting Figure 1 and 3 next to each other. They have too much overlap to be combined in one figure, but being able to compare the assimilation and validation points in a Figure 1a and 1b pair would be useful. Readers will surely be curious about which CSO measurements were assimilated, and at least the reason for not giving this information should be given in the paper.

**Response: This is a request from several reviewers and we are pleased to accommodate this request. We combined the two figures in the current draft of the manuscript.**

Reviewer #1 also asked for a comparison of estimated and measured snow densities. In addition to the insertion of rmse and bias statistics for SWE conversion from snow depth, it would be useful to state the mean SWE here for context.

**Response: The fieldwork measurement mean SWE is 51 cm and this statistic has been added to the final sentence in Section 6.4 (lines 584-585):**

**"The fieldwork measured mean SWE is 51 cm, the RMSE in SWE is 10.5 cm, and the Bias in SWE is 0.6 cm when using the Hill method for all fieldwork sites."**

Reviewer #2 asked for a more complete description of the DA approach. A crucial piece of information that is missing in the authors' response to this is that SnowAssim applies corrections retroactively, so simulations also change before the observation time. Readers could learn that by reading Liston and Hiemstra (2008) or figure it out by puzzling over Figure 5 (it is more obvious from Figure 6), but they could be spared that effort.

**Response: The authors agree that a clarification should be added to the manuscript regarding the retroactive corrections by SnowAssim. We adjusted several sentences in the SnowAssim method review in Section 3.2.4 (lines 208-212).**

"SnowAssim requires the model to be run twice and pauses at the end of the first model run. During this pause, differences between the observed SWE depths and modeled SWE depths in time and location are calculated and interpolated to the entire model domain in the form of a correction surface. The final correction surface is spatially distributed (for each day of observations) using the Barnes interpolation scheme. These correction surfaces are then applied to the precipitation inputs and snowmelt factors during the second model run."